# BEYOND THE PROXY: TRAJECTORY-DISTILLED GUIDANCE FOR OFFLINE GFLOWNET TRAINING

## ABSTRACT

Generative Flow Networks (GFlowNets) are effective at sampling diverse, high-reward objects, but in many real-world settings where new reward queries are infeasible, they must be trained from offline datasets. The prevailing training methods rely on a proxy model to provide reward feedback for online sampled trajectories. However, in scenarios where constructing a reliable proxy is challenging due to data scarcity or cost, one must turn to static offline trajectories for training. Nevertheless, current proxy-free approaches often rely on coarse constraints that may limit the model's ability to explore. To overcome these challenges, we propose **Trajectory-Distilled GFlowNet (TD-GFN)**, a novel proxy-free training framework. TD-GFN learns dense, transition-level edge rewards from offline trajectories via inverse reinforcement learning to provide rich structural guidance for efficient exploration. Crucially, to ensure robustness, these rewards are used indirectly to guide the policy through DAG pruning and prioritized backward sampling of training trajectories. This ensures that final gradient updates depend only on ground-truth terminal rewards from the dataset, thereby preventing the error propagation. Experiments show that TD-GFN significantly outperforms a broad range of existing baselines in both convergence speed and final sample quality, establishing a more robust and efficient paradigm for offline GFlowNet training.

## 1 INTRODUCTION

Generative Flow Networks (GFlowNets or GFNs) (Bengio et al., 2021; 2023) are a promising class of generative models designed to sample compositional objects in proportion to their positive rewards. They have demonstrated strong empirical success across diverse domains, including molecule discovery (Bengio et al., 2021), biological sequence design (Jain et al., 2022), combinatorial optimization (Zhang et al., 2025a), and the fine-tuning of generative models (Hu et al., 2024a; Liu et al., 2025). Despite their potential, in many real-world applications, interacting with the environment to obtain reward feedback is often impractical or infeasible due to costly experiments or the need for human evaluation. This necessitates learning GFlowNets from historically collected datasets.

The current prevailing approach involves training a proxy model on the dataset to approximate the reward signal, which the GFlowNet then queries during training to obtain rewards for the

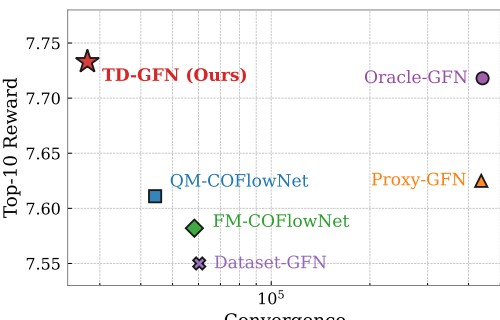

Figure 1: Comparison of methods on the Molecule Design task using a $1,500$-trajectory dataset. *Top-10 Reward* is the average of the 10 best samples, and *Convergence* is the number of training trajectories required for convergence. We compare against two proxy-based baselines from Bengio et al. (2021): Proxy-GFN (using a proxy learned on this dataset) and Oracle-GFN (using the oracle as a best-case proxy). See Section 4.3 for details.

terminal states and update itself. However, constructing a high-quality reward model is resource-intensive, typically demanding a large and diverse dataset along with substantial domain-specific expertise (Ouyang et al., 2022; Jain et al., 2023a). Moreover, queries for samples that are out-of-distribution for the proxy, or those it estimates inaccurately, can propagate errors directly through

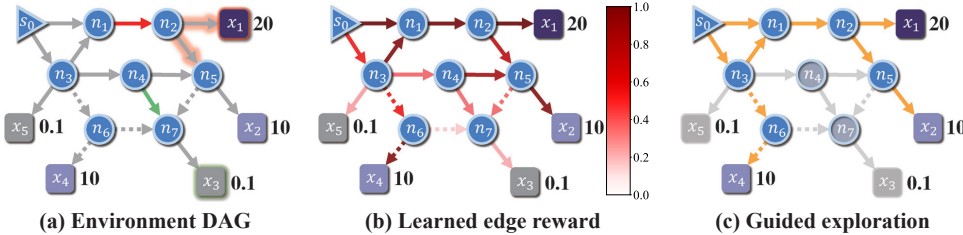

Figure 2: Illustration of our motivation. All subfigures depict an environment DAG, where solid edges represent transitions observed in the dataset and dashed edges indicate unobserved transitions. Reward values are annotated next to the terminal states. (a) Different edges contribute unequally to the final learned distribution. (b) Edge contributions as inferred by TD-GFN. (c) Selective exploration guided by the learned edge contributions.

gradients and corrupt the policy. These challenges are particularly pronounced in domains such as language modeling (Ziegler et al., 2019; Dalrymple et al., 2024), recommendation systems (Chen et al., 2024) and autonomous driving (Xie et al., 2024), where acquiring ground-truth rewards is often prohibitively expensive due to reliance on expert annotation or high trial-and-error costs.

Conversely, execution trajectories in these domains are inherently generated and readily available. While often overlooked in proxy-based methods, this trajectory-level information is a cornerstone of offline reinforcement learning (Kumar et al., 2020; Kostrikov et al., 2021) that has yet to be fully exploited for GFlowNets. As recent works expand GFlowNets into a broader spectrum of domains (Liu et al., 2023; Hu et al., 2024a; Zhu et al., 2025) to enable diverse generation, a critical question emerges: how can we effectively leverage these offline trajectories to train GFlowNets when the available terminal rewards are insufficient to construct a reliable proxy model?

Some recent efforts, such as RO-GFlowNets (Wang et al., 2023) and COFlowNet (Zhang et al., 2025b), have begun to explore learning directly from offline trajectories without reliance on proxy models. By leveraging these complete, pre-collected paths rather than relying on trial-and-error exploration, these methods converge to their optimal performance significantly faster during training, as illustrated in Figure 1. However, they typically impose coarse constraints to align the policy with the dataset, a practice that can hinder generalization, lead to poor exploration, and is highly sensitive to data quality.

In contrast to imposing such naive constraints, we introduce **Trajectory-Distilled GFlowNet (TD-GFN)**, a novel *proxy-free* framework that learns to extract rich structural information on the environment's directed acyclic graph (DAG) to guide its policy. Our work is built on an insight that different edges in the DAG are not equally important for learning an effective GFlowNet policy (also noted in prior work, e.g., Silva et al. (2025), though in a different setting). For instance, as shown in Figure 2(a), removing the red edge $(n_1 \to n_2)$ blocks access to a terminal node with reward 20, while removing the green edge $(n_4 \to n_7)$ only affects a low-reward node (0.1). To quantify this varying importance of edges, we employ inverse reinforcement learning (IRL) on a rebalanced offline dataset, leveraging the theoretical connection between GFlowNet training and reinforcement learning (Tiapkin et al., 2024) to distill transition-level scores we term edge rewards, as illustrated in Figure 2(b). These learned rewards provide dense, intermediate guidance that enables the policy to generalize across the DAG, fundamentally differing from a proxy model in both its learning objective and its method of application.

Crucially, to ensure robustness, we use these edge rewards in an effective *indirect* manner. First, we prune low-utility transitions as determined by their edge rewards, yielding a more compact and expressive action space that prioritizes high-value trajectories (Figure 2(c)). Second, we introduce a prioritized backward sampling procedure guided by both terminal and edge rewards, which strategically allocates the model's attention across both terminal states and intermediate pathways during training. Throughout this process, the gradient-based updates to our policy rely only on the ground-truth terminal rewards from the dataset. This indirect use of the learned edge rewards insulates the policy from potential inaccuracies in the IRL phase, preventing the error propagation that plagues proxy-based methods. More importantly, the dense reward guidance throughout its decision-making process enables TD-GFN to explore high-reward samples with superior efficiency. As shown in Figure 1, it achieves performance remarkably competitive with the Oracle-GFN, a GFlowNet trained

on a best-case proxy (pre-trained on $300k$ molecules) from Bengio et al. (2021), while using 20 times fewer training trajectories.

Extensive experiments demonstrate that our proposed TD-GFN trains with superior efficiency and reliability, significantly outperforming a wide range of baselines in convergence speed and final sample quality. These results establish TD-GFN as a more robust, stable, and efficient paradigm for offline GFlowNet training, marking a significant step forward from conventional methods that unlocks the potential for deploying GFlowNets in a broader spectrum of complex domains and scenarios.

In summary, our key contributions are as follows:

- **A Novel Proxy-Free Paradigm:** We introduce TD-GFN, a new paradigm for training GFlowNets from offline trajectory data that eliminates reliance on proxy models by avoiding out-of-distribution reward queries. It employs IRL to learn an edge reward for each transition on the DAG from a rebalanced offline dataset, which is then used to guide the policy's training and generalization across the environment.

- **Robust Guidance Mechanism:** We propose two mechanisms for indirectly guiding the policy using the learned edge rewards. Through pruning the DAG and prioritized backward sampling of training trajectories based on these edge rewards, we enable the model to train efficiently under dense guidance and ensure that the final gradient updates rely only on ground-truth rewards, thereby preventing the error propagation.

- **State-of-the-Art Performance and Efficiency:** Through extensive experiments, we demonstrate that TD-GFN establishes a new state of the art for training GFlowNets directly from offline trajectories. It significantly outperforms a wide range of baselines by robustly achieving a better fit to the target distribution, faster convergence, and higher sample quality.

## 2 PRELIMINARIES

In this section, we describe Generative Flow Networks (GFlowNets or GFNs) and the maximum causal entropy inverse reinforcement learning (IRL) framework, which help clarify our proposed method. Additional related works are provided in Appendix B for completeness.

### 2.1 GFLOWNETS

GFlowNets (Bengio et al., 2021; 2023) are defined within an environment represented as a directed acyclic graph (DAG), denoted by $G = (V, E)$, where $V$ is the set of nodes and $E$ the set of directed edges. A node $s'$ is said to be a child of node $s$ if and only if there exists a directed edge $(s \rightarrow s') \in E$; correspondingly, $s$ is the parent of $s'$. The graph includes a unique root node $s_0 \in V$ with no incoming edges and a subset of terminal nodes $\mathcal{X} \subseteq V$ with no outgoing edges.

A positive reward function $R(x)$ is defined over the terminal nodes $x \in \mathcal{X}$. The objective of GFlowNets is to learn a stochastic forward policy $\mathcal{P}_F(s'|s)$ such that the probability of reaching a terminal node $x \in \mathcal{X}$ is proportional to its associated reward $R(x)$. This is accomplished by generating sequences of transitions, where at each step, the agent selects a child node of the current state. In this work, we focus on deterministic environment dynamics, where each action uniquely determines the next state by traversing the corresponding edge in the DAG, as is standard in most existing GFlowNet studies (Jain et al., 2022; Jang et al., 2023; Hu et al., 2024a; Zhang et al., 2025a). Recent work (Tiapkin et al., 2024) establishes a formal equivalence between GFlowNet training and entropy-regularized RL under specific settings: a discount factor $\gamma = 1$, an entropy regularization coefficient $\lambda = 1$, and a modified reward function defined by a fixed backward policy $\mathcal{P}_B(s|s')$ that samples parent nodes in the DAG in reverse from a given child node:

$$\widetilde{R}(s, s') = \begin{cases} \log \mathcal{P}_B(s|s'), & \text{if } s \notin \mathcal{X} \cup \{s_f\}, \\ \log R(s), & \text{if } s \in \mathcal{X}, \\ 0, & \text{if } s = s_f, \end{cases} \tag{1}$$

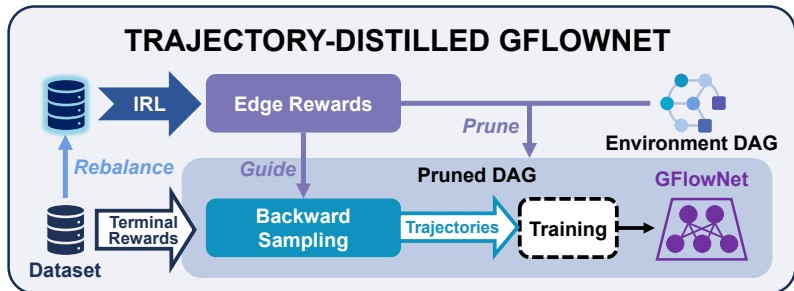

Figure 3: An overview of the complete training pipeline of the TD-GFN framework.

where $s_f$ denotes an additional absorbing state. Consequently, learning the forward policy reduces to solving the following entropy-regularized policy optimization problem:

$$\mathcal{P}_F^* = \arg\max_\pi \left( \lambda \mathcal{H}(\pi) + \mathbb{E}_\pi \left[ \sum_{t=0}^\infty \gamma^t \widetilde{R}(s_t, s_{t+1}) \right] \right), \qquad (2)$$

where $\mathcal{H}(\pi) = \mathbb{E}_\pi \left[ -\log \pi(s'|s) \right]$ denotes the causal entropy of the policy. The expectation $\mathbb{E}_\pi$ is taken with respect to trajectories generated by $s_0 \sim P_0$ and $s_{t+1} \sim \pi(\cdot|s_t)$, where $P_0$ is the initial state distribution that corresponds to a Dirac delta distribution in the GFlowNet setting.

In this paper, we address the problem of training a GFlowNet using an offline dataset $\mathcal{D} = \{\tau_i = (s_0, s_1, \ldots, s_{T_i}, R(s_{T_i}))\}_{i=1}^M$, which consists of trajectories of constructed objects generated by an unknown behavior policy, along with their corresponding ground-truth rewards.

## 2.2 MAXIMUM CAUSAL ENTROPY IRL

Maximum causal entropy inverse reinforcement learning (Ziebart et al., 2008; 2010) seeks to recover a reward function $r(s, s')$ that both explains expert behavior under a policy $\pi_E$ and promotes maximum causal entropy in the learned policy. This objective is formalized as:

$$\text{minimize}_r \left( \max_\pi \left[ \lambda \mathcal{H}(\pi) + \mathbb{E}_\pi \left( \sum_{t=0}^\infty \gamma^t r(s_t, s_{t+1}) \right) \right] \right) - \mathbb{E}_{\pi_E} \left[ \sum_{t=0}^\infty \gamma^t r(s_t, s_{t+1}) \right], \qquad (3)$$

where $\mathcal{H}(\pi)$ denotes the causal entropy of the policy $\pi$, consistent with its definition in Equation (2). In our approach, we adapt this framework to infer an edge-level reward function $R_E : E \to \mathbb{R}$ that reflects the expert policy's preferences for allocating attention across specific edges in the graph.

## 3 METHOD

In this section, we introduce **Trajectory-Distilled GFlowNet (TD-GFN)**, a novel proxy-free training framework. Unlike prior methods that constrain the policy to remain close to the dataset (Wang et al., 2023; Zhang et al., 2025b), TD-GFN achieves a more effective exploration–exploitation trade-off by strategically allocating attention across the environment DAG.

The overall training pipeline is illustrated in Figure 3. First, TD-GFN learns an *edge reward function* from a rebalanced dataset using inverse reinforcement learning (IRL). This function captures the relative importance of each edge in the environment DAG, providing principled guidance for policy learning. Leveraging these edge-level rewards, TD-GFN then prunes the DAG to remove low-utility transitions and trains the policy via prioritized backward sampling over the resulting subgraph. This design enables strategic generalization to broad, unseen regions of the state space while maintaining alignment with high-reward behaviors. The following subsections detail each component. A comprehensive summary of the full procedure is presented in Algorithm 1 (Appendix A).

## 3.1 EDGE REWARDS EXTRACTION VIA IRL

Motivated by the insight that different edges in the environment DAG contribute *unequally* to effective policy learning, we aim to quantify this disparity to enable more targeted and generalizable

training. Nevertheless, measuring the contribution of individual transitions under the inherently stochastic nature of GFlowNet policies poses a fundamental challenge. To tackle this, we leverage the theoretical equivalence between GFlowNet training and entropy-regularized reinforcement learning (Equation (2)) and applying the maximum causal entropy IRL framework (Ziebart et al., 2008; 2010) (Equation (3)) to extract a fine-grained *edge-level reward function* from offline trajectories. These learned edge rewards act as structured auxiliary signals that capture the relative importance of transitions, guiding the policy to generalize effectively across the environment DAG.

A central obstacle in applying IRL to offline data, however, lies in the fact that the behavior policy underlying the dataset may not reflect expert behavior. To mitigate this, we introduce a rebalancing strategy inspired by Hong et al. (2023) that biases trajectory sampling according to the reward of terminal states. Specifically, when estimating expectations under the expert policy $\mathbb{E}_{\pi_E}$, we sample trajectories $\tau = (s_0, s_1, \ldots, s_T) \in \mathcal{D}$ with probability $P(\tau) \propto R(s_T)$, thereby constructing a *rebalanced dataset* that more closely approximates the visitation distribution of an expert policy. This reweighting scheme effectively creates a pseudo-expert dataset by adjusting edge visitation frequencies to favor high-reward regions, thereby aligning the learned edge rewards with expert behavior more closely.

Inspired by the well-known maximum causal entropy IRL framework GAIL (Ho & Ermon, 2016) which has been theoretically and empirically shown to effectively learn reward functions with strong generalization (Xu et al., 2020; Luo et al., 2022; 2024), we adopt the following adversarial training objective based on the rebalanced dataset:

$$\underset{\phi}{\text{minimize}} \ \underset{\psi}{\max} \ \mathcal{L}(\psi, \phi) =$$
$$\left( \lambda \mathcal{H}(\pi_\psi) + \mathbb{E}_{s \sim \widetilde{\mathcal{D}}, s' \sim \pi_\psi(\cdot|s)} \left[ \log D_\phi(s, s') \right] \right) + \mathbb{E}_{(s,s') \sim \widetilde{\mathcal{D}}} \left[ \log \left( 1 - D_\phi(s, s') \right) \right],$$
(4)

where $\widetilde{\mathcal{D}}$ denotes both the node and edge distributions (with slight abuse of notation), $\pi_\psi$ is a parameterized policy and $D_\phi : E \to (0, 1)$ is a discriminative classifier, both optimized iteratively.

Nevertheless, the original GAIL framework induces a reward function that is strictly non-negative, making it inherently incapable of recovering the true underlying reward in our setting, which may include negative values (Equation (1)). To address this limitation, we follow Kostrikov et al. (2018) and extract an unbiased edge-level reward function $R_E$ from the classifier:

$$R_E(s, s') = \log D_\phi(s, s') - \log \left( 1 - D_\phi(s, s') \right)$$
(5)

Intuitively, the learned edge reward reflects the preference of the near-expert behavior policy underlying the rebalanced dataset, thereby quantifying the edge's importance for learning a high-quality GFlowNet policy. Fundamentally, this differs from proxy reward models: it is not designed to predict the reward of each terminal state, but rather to provide indirect, non-gradient guidance to the policy, as we demonstrate in the following section. The complete edge reward extraction procedure is summarized in Algorithm 2 in Appendix A.

As shown in Figure 2(b) and Appendix D.4, the learned edge rewards exhibit strong generalization, assigning meaningful values even to transitions not observed in the dataset. This guides the policy to allocate attention across the entire environment DAG, rather than staying confined to regions near the training data, as in prior methods. Crucially, this generalization capability is vital for the GFlowNet objective, as covering the target distribution inherently requires the policy to navigate and evaluate transitions in regions not explicitly covered by the offline dataset. Notably, such edge-level guidance across the entire DAG aligns more closely with the GFlowNet philosophy: leveraging structured, multi-step decision-making to decompose the complex problem of sampling from a target distribution, rather than attempting to model the distribution directly.

### 3.2 REWARD-GUIDED PRUNING AND PRIORITIZED BACKWARD SAMPLING

Although the learned edge reward provides a meaningful signal at the transition level, using it to directly shape the policy's gradients introduces sensitivity to function approximation errors—mirroring the error propagation commonly observed with proxy reward models. To enhance both the robustness and efficiency of the algorithm, we propose two novel mechanisms for shaping the GFlowNet policy as illustrated in Figure 3: we perform *reward-guided pruning* on the environment DAG and

subsequently train the GFlowNet using trajectories obtained via *prioritized backward sampling* on the pruned graph.

Guided by the learned edge reward, TD-GFN removes low-utility edges from the environment DAG $G = (V, E)$. Specifically, we approximate the distribution of edge rewards using a batch $\mathcal{D}_{R_E} = \{R_E(s_i, \text{selected}(\pi_\psi(\cdot \mid s_i)))\}_{i=1}^{B}$, where $\pi_\psi$ denotes the imitation policy obtained during edge reward learning (Algorithm 2), serving as a surrogate for expert behavior. An edge $(s \to s')$ is pruned from the graph if it lies in the low-density region of the reward distribution:

$$R_E(s, s') < \text{mean}(\mathcal{D}_{R_E}) - K \cdot \text{std}(\mathcal{D}_{R_E}), \tag{6}$$

where $K$ is a pruning threshold hyperparameter (a sensitivity analysis is provided in Appendix D.5), and $\text{mean}(\cdot)$, $\text{std}(\cdot)$ denote the empirical mean and standard deviation over the sampled batch $\mathcal{D}_{R_E}$. By adopting this threshold-based pruning strategy, we rely primarily on the model's ability to distinguish low-reward edges from potentially useful ones, thereby mitigating the risk of policy degradation caused by potential numerical errors in the learned IRL rewards. We further discuss alternative "soft" guidance mechanisms in Appendix E.

After applying this criterion, we further remove all edges that are disconnected from the root node $s_0$, resulting in a pruned subgraph $G' = (V', E')$ and an updated set of terminal nodes $\mathcal{X}'$. Notably, pruning is conducted over the full environment DAG, rather than being limited to dataset-observed transitions, enabling the retention of transitions that, while unobserved in the dataset, may potentially lead to high-reward regions.

This pruning approach concentrates learning on informative areas of the state space and improves training efficiency by reducing the complexity of model fitting (Zahavy et al., 2018; Zhang et al., 2020). The resulting subgraph defines a more compact yet expressive action space that prioritizes structurally meaningful and high-reward trajectories, thereby supporting policy generalization beyond observed data while remaining anchored in reliable behavioral signals, as illustrated in Figure 2(c).

To train the policy more efficiently on the pruned graph, we additionally propose a prioritized backward sampling mechanism that constructs training trajectories backwards from terminal nodes. Terminal states are sampled from the intersection of $\mathcal{X}'$ and dataset-supported objects $\{s_{T_i}\}_{i=1}^{M}$, with probability proportional to their known rewards. From each sampled terminal node $x$, we recursively perform backward sampling toward the root $s_0$ using a learned backward policy $\mathcal{P}_B$, defined as:

$$\mathcal{P}_B(s_t|s_{t+1}) = \frac{\exp\{R_E(s_t, s_{t+1})\}}{\sum_{(s, s_{t+1}) \in E'} \exp\{R_E(s, s_{t+1})\}}, \tag{7}$$

This formulation is consistent with the reward shaping principles in Equation (1). By sharpening policy updates toward regions prioritized by both terminal and edge rewards, this strategy reinforces the GFlowNet inductive bias of allocating sampling effort in proportion to reward, thereby improving sample efficiency. Moreover, when combined with the pruning method described above, it helps mitigate the under-exploitation of terminal nodes discussed in Jang et al. (2024).

Finally, we optimize the policy on the pruned DAG using the sampled trajectories and their corresponding terminal rewards from the offline dataset. Crucially, the gradient-based updates depend only on these recorded, ground-truth values. This approach not only sidesteps the fitting errors inherent in proxy reward models but also insulates the gradients from any potential inaccuracies in the edge rewards learned during the IRL phase.

By leveraging reward-guided pruning to steer the policy away from low-utility regions, and employing prioritized backward sampling to strategically allocate the model's attention based on rewards, we effectively achieve **exploration of unobserved high-utility regions during the evaluation phase**—a paradigm fundamentally distinct from the trial-and-error exploration inherent in proxy-based methods. Notably, the contributions of TD-GFN are orthogonal to the specific training strategies employed after trajectory sampling, as discussed in Appendix D.8.

## 4 EXPERIMENTS

In this section, we empirically demonstrate the effectiveness of the proposed TD-GFN through extensive experiments. Specifically, we aim to answer the following questions: (i) Can TD-GFN

effectively sample from the target reward distribution, and generate high-reward and diverse samples? (ii) Is TD-GFN training efficient? (iii) Does TD-GFN consistently improve performance across different scenarios? We also conduct ablation studies to evaluate the individual contributions of each component within our method, as presented in Appendix D.7. Comprehensive task descriptions and implementation details of the algorithm are provided in Appendix C. Further experimental results on additional real-world datasets can be found in Appendix D.1.

### 4.1 HYPERGRID

We begin with experiments on the Hypergrid task (Bengio et al., 2021), a $D$-dimensional grid environment with side length $H$, resulting in a discrete space of size $H^D$. High-reward modes are narrowly concentrated near the $2^D$ corners, making the task particularly challenging for both exploration-based and offline learning algorithms. Our main experiments are conducted on a $8^4$ Hypergrid ($H = 8$, $D = 4$) following the setup in Bengio et al. (2021). We also provided results on larger Hypergrids in Appendix D.3. Additionally, in Appendix D.4, we visualize a $8 \times 8$ Hypergrid instance and illustrate the TD-GFN workflow on this task.

Due to the nascent nature of proxy-free GFlowNet training, we compare TD-GFN against a broad range of baselines. These include the main existing proxy-free GFlowNet method, COFlowNet (Zhang et al., 2025b), as well as popular offline RL algorithms (soft RL version) such as CQL (Kumar et al., 2020) and IQL (Kostrikov et al., 2021), imitation learning methods such as Behavior Cloning (BC) and GAIL (Ho & Ermon, 2016). We additionally include a baseline termed *Dataset-GFN*, which trains a GFlowNet directly on offline trajectories without modification, serving as a vanilla proxy-free variant. To ensure a fair comparison with COFlowNet, which is based on the flow matching (FM) objective (Bengio et al., 2021), we also adopt the FM objective (Equation (8)) in our experiments. Results for alternative objectives applied directly to offline trajectories are provided in Appendix D.12. Likewise, the following experiments are conducted under this same setting (results with additional objectives can be found in Appendix D.8). Moreover, we train GAIL using the rebalanced dataset described in Section 3.1 (rebalanced GAIL), which improves its performance relative to training on the original dataset. Accordingly, we evaluate it using the imitation policy learned via Algorithm 2.

Each algorithm is evaluated using two standard metrics (Bengio et al., 2021; Pan et al., 2022; Zhang et al., 2023b; 2025b). The first is *Empirical L1 Error*, which measures the $\mathcal{L}_1$ distance between the empirical distribution $\pi(x)$ from model samples and the normalized reward distribution $p(x) = R(x)/Z$, computed as $\mathbb{E}[|\pi(x) - p(x)|]$. The second is *Modes Found*, defined as the number of distinct reward modes discovered during training.

We construct an *Expert* dataset of $1,500$ trajectories using a well-trained GFlowNet policy, simulating data that would otherwise be collected by a domain expert. It is important to clarify that we adopt GFlowNet-generated datasets not because our approach requires them, but because they provide a controlled and reproducible way to simulate realistic scenarios in which data arises from behavior policies of different optimality levels. This strategy is also common in the offline reinforcement learning community (Fu et al., 2020; Gulcehre et al., 2020). As demonstrated in the Section 4.2, TD-GFN also delivers compelling results on a real-world dataset. A more detailed discussion on this topic, as well as an analysis of trajectory structure's value, is provided in Appendix E.

As shown in Figure 4(a), TD-GFN demonstrates superior sample efficiency. It identifies all 16 modes with under $5,000$ state visits—a **six-fold** improvement over baselines—and converges to the lowest *Empirical L1 Error* more than twice as fast as the next best method, COFlowNet. A complete analysis of the other baselines can be found in Appendix D.9.

**Robustness to Dataset Uncertainty.** We first assess the robustness of TD-GFN under dataset uncertainty by introducing greater variability in the training data. In the *Mixed* setting, we augment the *Expert* dataset with an equal number of trajectories generated by a random policy similar to Hong et al. (2023); Cao et al. (2024), thereby introducing noise. In another setting, we significantly reduce the number of available trajectories—using only one-tenth of the original dataset—to simulate data scarcity as an additional source of uncertainty (Depeweg et al., 2018).

As shown in Figure 4, TD-GFN performs robustly under both types of uncertainty. It consistently models the target distribution more accurately and finds high-reward modes significantly faster than

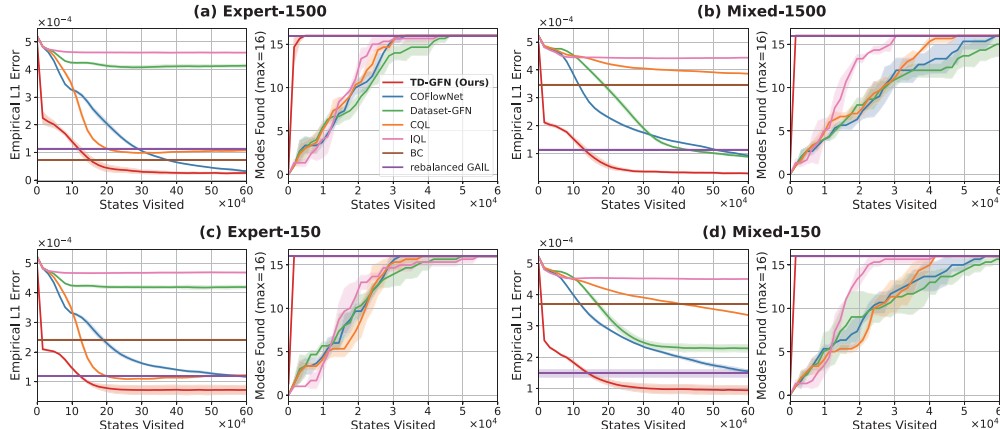

Figure 4: Performance comparison on the $8^4$ Hypergrid task. The top row shows results using $1,500$ trajectories from the *Expert* and *Mixed* datasets for policy training, while the bottom row shows results using only $150$ trajectories. The solid line and shaded region represent the mean and standard deviation, respectively; the same convention applies to the following figures.

.

baseline methods. These results highlight TD-GFN's resilience to noisy and limited data. Further evaluation under extreme data scarcity appears in Appendix D.2.

**Robustness to Behavior Policy Quality.**    We also examine the sensitivity of TD-GFN to the quality of the behavior policy used during data collection. To this end, we construct two offline datasets. The first, denoted *Median*, is collected using a suboptimal GFlowNet trained for only half the number of steps required for convergence. The second, denoted *Bad*, is generated by a GFlowNet trained with an inverted reward function $IR = R^{-0.1}$, while the dataset still contains the true rewards.

As shown in Figure 5, TD-GFN adapts effectively in both scenarios, achieving faster alignment with the target reward distribution and recovering all reward modes earlier than baseline methods. Notably, when the behavior policy diverges substantially from the expert (*Bad*), TD-GFN exhibits significantly more accurate distribution modeling, underscoring its robustness under adverse conditions.

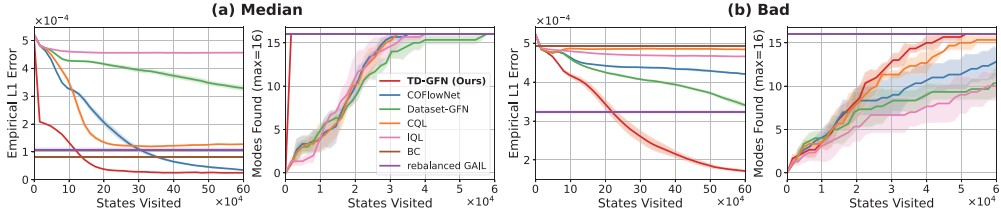

Figure 5: Performance comparison on the $8^4$ Hypergrid task. Policies are trained using the *Median* and *Bad* datasets, each consisting of $1,500$ trajectories.

Notably, the sampled $150$ trajectories from the *Mixed* dataset and $1,500$ trajectories from the *Bad* dataset cover only **12** and **6** out of 16 reward modes, respectively. This confirms that our experiments effectively evaluate the method's capacity to generate novel states unobserved in the dataset, rather than merely assessing its degree of overfitting.

## 4.2 BIOSEQUENCE DESIGN

We next evaluate TD-GFN on a more challenging task—designing anti-microbial peptides (AMPs), i.e., short protein sequences—introduced in Jain et al. (2022). The experimental setup leverages two datasets curated from the DBAASP database (Pirtskhalava et al., 2021): one for training a reward proxy and another for an oracle that provides ground-truth labels.

Using the proxy reward model provided in Jain et al. (2022), we train a Proxy-GFN and compare its performance with TD-GFN, as well as with other baseline methods previously evaluated on the

Hypergrid task (Section 4.1). All methods are trained on the same dataset used for proxy learning, which includes $3,219$ positive AMPs and $4,761$ non-AMPs.

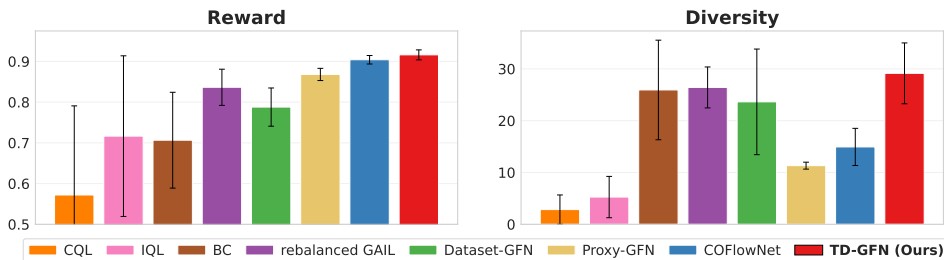

Figure 6: Comparison of reward and diversity among the top $100$ sequences ranked by reward. The *Diversity* metric is defined in Appendix C.2. All results are reported over three random seeds.

We generate $5,000$ sequences using each learned policy and report the reward and diversity metrics for the top $100$ sequences ranked by reward, as shown in Figure 6. TD-GFN consistently produces sequences with the highest rewards and greatest diversity. Notably, although rebalanced GAIL merely imitates the policy implied by the rebalanced dataset, it still achieves strong performance—outperforming Dataset-GFN, which is trained directly on the dataset trajectories, in both reward and diversity. These findings highlight the effectiveness of our dataset rebalancing strategy.

## 4.3  MOLECULE DESIGN

We further evaluate our method on the real-world molecular design benchmark introduced by Bengio et al. (2021). We use a training dataset that simulates historical expert data, consisting of $1,500$ trajectories generated by a moderately-trained GFlowNet policy (Behavior-GFN) and rewarded by the oracle provided in the original paper which was pre-trained on $300k$ molecules.

Table 1: Comparison of methods on the Molecule Design task. The *Convergence* column indicates the number of trajectories required for performance convergence. Results are reported as mean $\pm$ standard deviation over three random seeds, and bold numbers indicate the best performance in each column. The "Dataset" row shows the information of the $1,500$ samples in the training dataset for reference.

| Method | Reward-10 ($\uparrow$) | Reward-100 ($\uparrow$) | Reward-1000 ($\uparrow$) | Convergence ($\downarrow$) |
|---|---|---|---|---|
| Dataset | $7.420 \pm 0.088$ | $6.968 \pm 0.226$ | $5.757 \pm 0.635$ | / |
| Behavior-GFN | $7.534 \pm 0.066$ | $7.220 \pm 0.165$ | $6.504 \pm 0.348$ | / |
| Oracle-GFN | $7.718 \pm 0.014$ | $7.408 \pm 0.021$ | $6.801 \pm 0.023$ | $44.141 \times 10^4$ |
| CQL | $7.069$ | $6.643$ | $5.401$ | $0.803 \times 10^4$ |
| IQL | $6.902$ | $5.980$ | $4.628$ | $4.104 \times 10^4$ |
| BraVE | $7.271$ | $6.650$ | $5.590$ | $\mathbf{0.645 \times 10^4}$ |
| BC | $7.652 \pm 0.053$ | $7.223 \pm 0.035$ | $6.459 \pm 0.016$ | / |
| GAIL | $7.528 \pm 0.068$ | $7.152 \pm 0.085$ | $6.406 \pm 0.033$ | / |
| Proxy-GFN | $7.625 \pm 0.063$ | $7.281 \pm 0.067$ | $6.636 \pm 0.097$ | $43.735 \times 10^4$ |
| Dataset-GFN | $7.550 \pm 0.045$ | $7.198 \pm 0.018$ | $6.474 \pm 0.018$ | $6.030 \times 10^4$ |
| FM-COFlowNet | $7.582 \pm 0.057$ | $7.201 \pm 0.015$ | $6.485 \pm 0.016$ | $5.829 \times 10^4$ |
| QM-COFlowNet | $7.611 \pm 0.020$ | $7.296 \pm 0.022$ | $6.638 \pm 0.010$ | $4.423 \times 10^4$ |
| **TD-GFN (Ours)** | $\mathbf{7.733 \pm 0.036}$ | $\mathbf{7.450 \pm 0.037}$ | $\mathbf{6.810 \pm 0.035}$ | $\mathbf{2.749 \times 10^4}$ |

Following the architecture and methodology commonly adopted in prior GFlowNet research (Bengio et al., 2021; Malkin et al., 2022; Jang et al., 2023; Tiapkin et al., 2024), we train a proxy model on this dataset, which is then used to train a GFlowNet baseline referred to as Proxy-GFN. We also train an Oracle-GFN that uses the oracle model itself as the best-case proxy. We evaluate both the original (FM) COFlowNet and its quantile-augmented (QM) variant, proposed by Zhang et al. (2025b), which uses quantile matching (Zhang et al., 2023b) to improve candidate diversity in this task. We also

incorporate BraVE (Landers et al., 2025), a recent offline RL algorithm tailored for discrete action spaces, to broaden our comparison with non-GFlowNet methods.

We report the average top-$k$ reward obtained from $5,000$ samples generated by policies trained with different algorithms, along with the number of training trajectories required for performance convergence, in Table 1. Due to the instability of offline RL methods under sparse rewards, results for offline RL algorithms are reported using only their best-performing random seed. In Appendix D.10, we further report the Tanimoto similarity (Bajusz et al., 2015) among sampled molecules.

As shown in the table, BraVE converges fastest in terms of sample efficiency but struggles to generate a diverse set of high-reward candidates due to the fundamental misalignment between the standard RL paradigm and the task's diversity objective, resulting in substantially lower top-$k$ average rewards than GFlowNet-based and even imitation learning methods. Excluding offline RL methods, TD-GFN achieves the best performance in discovering high-reward molecules with the fewest trajectories. Additional comparisons of training time and computational resource usage are provided in Appendix D.6.

Notably, the powerful guidance provided at intermediate steps allows TD-GFN to match Oracle-GFN's performance using **20 times fewer** trajectories, while also decisively surpassing the weaker data-generating policy. This remarkable efficiency showcases its ability to transcend the limitation of training data and signals strong potential for iterative policy evolution.

To evaluate the diversity of molecules generated, we measure the number of distinct high-reward modes identified by each algorithm. Modes are clusters of high-reward molecules based on Tanimoto similarity (Bajusz et al., 2015). As shown

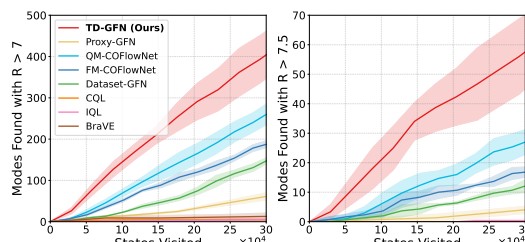

Figure 7: Number of high-reward modes (reward $> 7$ and $> 7.5$) discovered during training in the Molecule Design task.

in Figure 7, TD-GFN consistently discovers $1.5$ to $2$ times more high-reward modes than the strongest baselines, indicating superior exploration. Notably, despite incorporating DAG pruning, TD-GFN does not overfit to narrow regions of the search space. Instead, it guides the policy toward a broader set of high-reward regions, balancing targeted exploitation with diverse exploration.

## 5 CONCLUSION

We introduced **Trajectory-Distilled GFlowNet (TD-GFN)**, a proxy-free training framework for learning GFlowNets from offline trajectory datasets without requiring out-of-distribution reward queries. Leveraging edge-level guidance learned via IRL, TD-GFN consistently steers the policy toward improved performance and training efficiency through DAG pruning and prioritized backward trajectory sampling. Extensive experiments demonstrate state-of-the-art performance and efficiency.

## REPRODUCIBILITY STATEMENT

To ensure the reproducibility of our findings, we provide detailed information regarding our experimental setup. All model architectures, hyperparameter settings, and training configurations are thoroughly documented in Appendix C. The source code for our experiments will be made publicly available upon publication of this work.

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

# Supplementary Materials

## A ALGORITHM PSEUDOCODE

---

**Algorithm 1** Trajectory-Distilled GFlowNet (TD-GFN)

---

1: **Input:** Offline dataset $\mathcal{D} = \{\tau_i = (s_0, \ldots, s_{T_i}, R(s_{T_i}))\}_{i=1}^M$; number of training iterations $N$; batch size $B$; GFlowNet learning rate $\eta$.
2: **Phase 1: Edge Reward Extraction**
3: Learn edge rewards $R_E(s, s')$ and the imitation policy $\pi_\psi$ via Algorithm 2.
4: **Phase 2: DAG Pruning**
5: Collect edge rewards $\mathcal{D}_{R_E}$ using $R_E(s, s')$ and $\pi_\psi$.
6: Prune the environment DAG $G = (V, E)$ based on the thresholding rule defined in Equation (6).
7: Remove disconnected edges to obtain the pruned DAG $G' = (V', E')$.
8: **Phase 3: Policy Training**
9: Initialize the parameterized GFlowNet policy $\mathcal{P}_F$.
10: **for** $n = 1$ to $N$ **do**
11:     Initialize a batch of trajectories $\mathcal{T} = \varnothing$.
12:     **for** $j = 1$ to $B$ **do**
13:         Sample a terminal node $x_j \in \mathcal{X}' \cap \{s_{T_i}\}_{i=1}^M$ with probability $P(x) \propto R(x)$.
14:         Initialize a trajectory $\tau_j = \varnothing$, $s_{now} = x_j$, $s_{pre} = x_j$.
15:         **while** $s_{now} \neq s_0$ **do**
16:             Sample $s_{pre} \sim \mathcal{P}_B(\cdot|s_{now})$, where $\mathcal{P}_B$ is defined in Equation (7) over $G'$.
17:             $\tau_j = \tau_j \cup \{(s_{pre} \to s_{now})\}$.
18:             $s_{now} = s_{pre}$
19:         **end while** // Sample training trajectories via prioritized backward process.
20:         $\mathcal{T} = \mathcal{T} \cup \{\tau_j\}$.
21:     **end for**
22:     Optimize $\mathcal{P}_F$ over $G'$ with sampled trajectories $\mathcal{T}$. // Update policy.
23: **end for**
24: **Output:** Trained GFlowNet policy $\mathcal{P}_F$.

---

**Algorithm 2** Edge Reward Extraction

---

1: **Input:** Offline dataset $\mathcal{D} = \{\tau_i = (s_0, \ldots, s_{T_i}, R(s_{T_i}))\}_{i=1}^M$; number of training iterations $N'$; batch size $B'$; policy learning rate $\alpha$; discriminator learning rate $\beta$.
2: Initialize policy $\pi_\psi$ and discriminator $D_\phi : E \to (0, 1)$ with random parameters $\psi$ and $\phi$.
3: **for** $n = 1$ to $N'$ **do**
4:     Initialize a batch of trajectories $\mathcal{T}' = \varnothing$.
5:     **for** $j = 1$ to $B'$ **do**
6:         Sample a trajectory $\tau_j \in \mathcal{D}$ with probability $P(\tau_j) \propto R(s_{T_j})$ .
7:         $\mathcal{T}' = \mathcal{T}' \cup \{\tau_j\}$.
8:     **end for** // Sample from the rebalanced dataset.
9:     Extract edge transitions $\{(s, s')\}$ from $\mathcal{T}'$ as a minibatch $\overline{E}$.
10:     Compute $\nabla_\phi L$ on $\overline{E}$; update $\phi \leftarrow \phi - \beta \nabla_\phi L$. // Update discriminator.
11:     Compute $\nabla_\psi L$ on $\overline{E}$; update $\psi \leftarrow \psi - \alpha \nabla_\psi L$ or apply a policy gradient method. // Update imitation policy.
12: **end for**
13: Extract the edge reward function $R_E$ according to Equation (5).
14: **Output:** Edge reward function $R_E$; imitation policy $\pi_\psi$.

---

## B RELATED WORK

**Generative Flow Networks (GFlowNets).** GFlowNets (Bengio et al., 2021; 2023) are a class of generative models designed to sample compositional objects from an unnormalized reward distribution via sequential action selection. They have been theoretically linked to entropy-regularized reinforcement learning (Tiapkin et al., 2024; Mohammadpour et al., 2024) and variational inference (Malkin et al., 2023; Zimmermann et al., 2023), offering a principled framework for distribution matching.

Recent research has focused on improving GFlowNet training by proposing new balance conditions (Malkin et al., 2022; Madan et al., 2023), enhancing credit assignment through intermediate rewards (Pan et al., 2022; 2023a; Jang et al., 2023), designing backward policies for improved sampling (Jang et al., 2024; Mohammadpour et al., 2024; Shen et al., 2023; Gritsaev et al., 2024), refining sampling and resampling strategies (Atanackovic & Bengio, 2024; Kim et al., 2024c;b; Lau et al., 2024; Madan et al., 2025), and exploring alternative training objectives (Silva et al., 2024; Hu et al., 2024b), including energy-based training approaches for diffusion-structured inference (Campbell et al., 2024). Extensions to the GFlowNet framework have expanded its applicability to stochastic dynamics (Pan et al., 2023b), continuous action spaces (Lahlou et al., 2023), non-acyclic transitions (Brunswic et al., 2024; Morozov et al., 2025), and implicit reward feedback (Chen & Mauch, 2024). These advances have broadened the scope of GFlowNets, enabling impactful applications in molecule design (Jain et al., 2023a;b; Zhu et al., 2023; Roy et al., 2023), biological sequence generation (Jain et al., 2022; Ghari et al., 2023), combinatorial optimization (Zhang et al., 2025a; 2023a; Kim et al., 2024a), causal inference (Zhang et al., 2022; Deleu et al., 2022), recommendation systems (Liu et al., 2023), and fine-tuning of large language models (Hu et al., 2024a; Yu et al., 2024) and diffusion models (Liu et al., 2025; Venkatraman et al., 2024).

Our work is most closely related to recent efforts in offline and proxy-free GFlowNet training. An early approach, RO-GFlowNets (Wang et al., 2023), uses offline action probabilities to constrain the learned policy, while COFlowNet (Zhang et al., 2025b) penalizes flow on unseen edges to limit policy coverage to dataset-adjacent regions. Both methods aim to improve policy reliability by imposing coarse constraints that merely align policy behavior or coverage more closely with the dataset. However, such constraints impose blunt limitations that hinder generalization and make performance highly sensitive to data quality, as they fail to fully exploit the trajectory-level information contained in the dataset. Other related works include Atanackovic & Bengio (2024), which explores an "offline" setting but primarily focuses on evaluating sampling strategies, and the KL-weakFM loss (Brunswic et al., 2025), which enables proxy-free imitation learning. Concurrently, some research has investigated improving offline training with explicit reward signals and novel exploration strategies (Sendera et al., 2024), providing complementary perspectives to our pruning-based framework.

Conceptually, our work shares its core insight with Silva et al. (2025), which introduces Weighted Detailed Balance (WDB) to emphasize transitions with a larger impact on correctness, particularly in online GFlowNets with GNN-based parameterizations. Their analysis demonstrates that imbalances across edges can systematically affect sampling accuracy. Our work shares this motivation but takes a different route: instead of reweighting balance conditions, we employ inverse reinforcement learning (IRL) to estimate edge rewards. These rewards are then used to guide pruning and prioritized sampling in the offline setting. This distinction in methodology makes the two approaches complementary—WDB could potentially enhance offline training, while our pruning and sampling strategies may in turn benefit online methods.

**Inverse Reinforcement Learning (IRL).** Inverse reinforcement learning (IRL) aims to recover a reward function that explains observed expert behavior in a Markov decision process (Ng et al., 2000). Among its variants, maximum causal entropy IRL is particularly noteworthy, as it addresses reward ambiguity by formulating the objective as finding a policy that maximizes entropy while matching the observed expert behavior (Ziebart et al., 2008; 2010). Building on this framework, Generative Adversarial Imitation Learning (GAIL) (Ho & Ermon, 2016) reinterprets imitation as a distribution matching problem via adversarial training, where a discriminator distinguishes expert from policy-generated trajectories, and its output serves as a learned reward for policy training. GAIL eliminates the need to explicitly solve the inner IRL loop, significantly improving scalability, and has inspired a broad class of Adversarial Imitation Learning (AIL) methods (Fu et al., 2018; Kostrikov et al., 2018; 2020; Garg et al., 2021). This line of work has since been extended in various directions, including efforts to improve reward generalization and robustness (Xu et al., 2020; Luo et al., 2022; 2024).

## C  EXPERIMENTAL DETAILS

### C.1  HYPERGRID

The Hypergrid task is a synthetic benchmark introduced in Bengio et al. (2021) to evaluate the exploration and generalization capabilities of GFlowNet algorithms. It presents a significant challenge due to its high-dimensional, sparse reward landscape.

The environment consists of a $D$-dimensional hypercubic grid with side length $H$, defining a discrete state space of size $H^D$. Each state corresponds to a coordinate $x = (x_1, \ldots, x_D) \in \{0, \ldots, H-1\}^D$. The agent starts at the origin $x = (0, \ldots, 0)$ and can increment any individual coordinate by one at each step. A trajectory terminates when a special stop action is taken, and the reward is assigned based on the final state.

The reward function is designed to produce multiple sharp reward modes near the corners of the hypercube. Formally, the reward at a terminal state $x$ is defined as:

$$R(x) = R_0 + R_1 \prod_{d=1}^{D} \mathbb{I}\left(\left|\frac{x_d}{H} - 0.5\right| > 0.25\right) + R_2 \prod_{d=1}^{D} \mathbb{I}\left(0.3 < \left|\frac{x_d}{H} - 0.5\right| < 0.4\right),$$

where $R_0 < R_1 \ll R_2$, and $\mathbb{I}(\cdot)$ denotes the indicator function. The first term, weighted by $R_1$, introduces $2^D$ high-reward modes near the grid's corners, while the second term, weighted by $R_2$, creates even sparser regions of extremely high reward, thereby increasing the difficulty of discovering them through exploration alone.

Following the most challenging configuration proposed in Bengio et al. (2021), we set the environment parameters to $R_0 = 10^{-3}, R_1 = 0.5, R_2 = 2$. This configuration results in a highly multi-modal and sparsely rewarded environment, making it well-suited for evaluating the effectiveness of offline training strategies.

### C.2  BIOSEQUENCE DESIGN

The Anti-Microbial Peptide (AMP) Design task, introduced in Jain et al. (2022), is a realistic sequence generation benchmark aimed at designing short peptide sequences with strong anti-microbial properties.

The agent constructs sequences by selecting amino acids from a fixed vocabulary of 20 standard residues, with a maximum sequence length of 50. At each time step, the agent chooses either an amino acid or a special end-of-sequence token, proceeding in a left-to-right manner. As a result, the transition structure forms a tree, and the trajectories in offline datasets can be directly inferred by the terminal states.

Notably, when the environment DAG degenerates into a tree, the prioritized backward sampling strategy in TD-GFN simplifies to sampling trajectories from the dataset in proportion to their terminal rewards. Despite this simplification, TD-GFN remains effective on this task, which presents substantial challenges due to the large combinatorial space of valid sequences ($\approx 20^{50}$) and the extreme sparsity of high-reward regions.

In our experiments, we adopt the *Diversity* metric to evaluate algorithmic performance, following the methodology in Jain et al. (2022). For a set of biological sequences $C$, the metric is defined as:

$$\text{Diversity}(C) = \frac{\sum_{x_i, x_j \in C} \text{Lev}(x_i, x_j)}{|C|(|C| - 1)},$$

where $\text{Lev}(\cdot, \cdot)$ denotes the Levenshtein distance. All environmental parameters are kept consistent with those used in Jain et al. (2022).

### C.3  MOLECULE DESIGN

We evaluate our method on a fragment-based molecular generation task targeting protein binding affinity, specifically for soluble epoxide hydrolase (sEH). The goal is to construct candidate molecules with high predicted affinity for the target protein. Molecules are generated sequentially using the

junction tree framework introduced in Jin et al. (2018), where each action involves selecting an attachment atom and adding a fragment from a predefined vocabulary of 105 building blocks.

The reward is defined as the negative binding energy between the generated molecule and the target protein, as predicted by an oracle model pre-trained on $300k$ molecules, following the setup in Bengio et al. (2021). To balance reward maximization with output diversity, the final reward is modulated using a reward exponent $\omega$, such that $R'(x) = R(x)^{\omega}$, where $R(x)$ denotes the raw oracle prediction. All environmental parameters, including $\omega$, are kept consistent with those used in Bengio et al. (2021).

## C.4  POLICY MODEL ARCHITECTURES AND HYPERPARAMETERS

Across the experimental tasks presented in this work, we adopt the model architectures listed in Table 2 as the backbone structures for the policy networks.

Table 2: Policy network architectures used across different experimental tasks.

| Task | Model architecture |
|---|---|
| Hypergrid | MLP([input, 256, 256, output]) |
| Biosequence Design | MLP([input, 128, 128, output]) |
| Molecule Design | MPNN (Gilmer et al., 2017) v4 |

Table 3: Hyperparameters used across different tasks and methods.

| Method | Hyperparameter | Task | Value |
|---|---|---|---|
| All | Policy learning rate $\eta$ | Hypergrid | $1 \times 10^{-5}$ |
| | | Molecule Design | $1 \times 10^{-4}$ |
| | | Biosequence Design | $1 \times 10^{-3}$ |
| | Entropy coef. $\lambda$ | All | 0.01 |
| | Seeds | All | 0,1,2 |
| TD-GFN | Pruning coef. $K$ | Hypergrid | 7.0 |
| | | Molecule Design | 1.0 |
| | | Biosequence Design | 2.0 |
| | Actor learning rate $\alpha$ | Hypergrid | $1 \times 10^{-5}$ |
| | | Molecule Design | $1 \times 10^{-4}$ |
| | | Biosequence Design | $1 \times 10^{-3}$ |
| | Discriminator learning rate $\beta$ | Hypergrid | $3 \times 10^{-5}$ |
| | | Molecule Design | $3 \times 10^{-4}$ |
| | | Biosequence Design | $3 \times 10^{-3}$ |
| COFlowNet | Regularizing coef. 1 | Hypergrid | 1.0 |
| | | Molecule Design | 1.0 |
| | | Biosequence Design | / |
| | Regularizing coef. 2 | Hypergrid | 1.0 |
| | | Molecule Design | 10.0 |
| | | Biosequence Design | 25.0 |
| IQL | Expectile | Hypergrid | 0.9 |
| | | Molecule Design | 0.9 |
| | | Biosequence Design | 0.8 |
| BC | Loss | All | Cross Entropy Loss |

Key algorithmic hyperparameters used across different tasks and methods are summarized in Table 3. These values were selected through empirical tuning to ensure stable training and fair comparisons among baselines.[1] Specifically, the pruning threshold coefficient $K$ is tuned by selecting the value that yields the highest ratio of retained recorded transitions to retained randomly selected transitions. For the sensitivity analysis with respect to parameter $K$, please refer to Appendix D.5.

---

[1] Our implementation of COFlowNet is based on the official open-source repository available at `https://github.com/yuxuan9982/COflownet`.

# D ADDITIONAL EXPERIMENTAL RESULTS

## D.1 ADDITIONAL EXPERIMENTS ON REAL-WORLD RECOMMENDATION DATASETS

To further evaluate TD-GFN on a **real-world combinatorial problem modeled as a general DAG**, we introduce an evaluation on the listwise recommendation task following Liu et al. (2023).

**Environment** The environment is built upon the MovieLens 1M (ML1M) dataset[2], a standard and widely-used public benchmark for recommendation systems comprising real user-item interactions. The agent is tasked with autoregressively generating a list (or "slate") of $K = 6$ unique items for a user. With a total item pool of $3,706$, this presents a large-scale combinatorial generation challenge. Following Liu et al. (2023), the reward for a generated list $O$ is defined as the average of its item-wise rewards, $\mathcal{R}(u, O) = \frac{1}{K} \sum_{i \in O} \mathcal{R}(u, i)$, where item-wise rewards are mapped from user ratings (clicks, likes, stars) to a scale of $[0, 3]$.

Crucially, since the utility of a recommendation slate is invariant to the permutation of its items, any specific set of unique items can be generated via $K! = 720$ distinct sequential paths. This characteristic explicitly structures the environment as a non-degenerate DAG, distinguishing it from tree-structured tasks. For our offline setting, we filtered the public dataset to retain users with at least 25 interactions. We then constructed the training dataset by directly sampling trajectories of length $K = 6$ from these users' real-world interaction logs.

**Metrics** We evaluate all methods using three standard metrics from the Liu et al. (2023) benchmark:

- **Average Reward**: The mean listwise reward over a large batch of generated samples.
- **Max Reward**: The reward of the single best list generated.
- **Coverage**: The total number of unique items that appear across all generated lists, measuring sample diversity.

**Results** The performance of all methods on the offline ML1M dataset is presented in Table 4. TD-GFN consistently outperforms other offline GFN baselines; notably, it achieves reward performance competitive with the Oracle-GFN while exhibiting significantly higher and more stable diversity.

Table 4: Performance comparison on the ML1M listwise recommendation task. Results are reported as mean $\pm$ standard deviation over three random seeds.

| Method | Avg. Reward ($\uparrow$) | Max Reward ($\uparrow$) | Coverage ($\uparrow$) |
|---|---|---|---|
| Oracle-GFN | $\mathbf{2.092 \pm 0.033}$ | $\mathbf{2.930 \pm 0.017}$ | $113.7 \pm 72.0$ |
| Dataset-GFN | $1.681 \pm 0.202$ | $2.674 \pm 0.184$ | $7.9 \pm 0.9$ |
| QM-COFlowNet | $1.787 \pm 0.022$ | $2.843 \pm 0.020$ | $119.2 \pm 6.1$ |
| **TD-GFN (Ours)** | $\mathbf{1.988 \pm 0.006}$ | $\mathbf{2.914 \pm 0.013}$ | $\mathbf{186.8 \pm 6.3}$ |

## D.2 ALGORITHM TRAINED ON EXTREMELY LIMITED DATA

To further assess the robustness of TD-GFN under extreme data scarcity, we conduct experiments on the $H = 8$, $D = 4$ Hypergrid environment using only **30** training trajectories from the *Expert/Mixed* datasets which cover merely **5** and **3** modes respectively, as illustrated in Figure 8. In this setting, algorithms are prone to overfitting due to the limited data. For example, both COFlowNet and CQL exhibit non-monotonic trends in *Empirical L1 Error*, initially decreasing but later increasing during training, indicating overfitting to the limited known regions.

In contrast, TD-GFN consistently converges toward the target distribution throughout training. Owing to the strong generalization capabilities of the learned edge rewards, it avoids overfitting to the dataset and quickly identifies high-reward modes. These results highlight TD-GFN's ability to maintain

---

[2]https://grouplens.org/datasets/movielens/1m/

a favorable balance between exploration and exploitation, even under severely constrained data conditions.

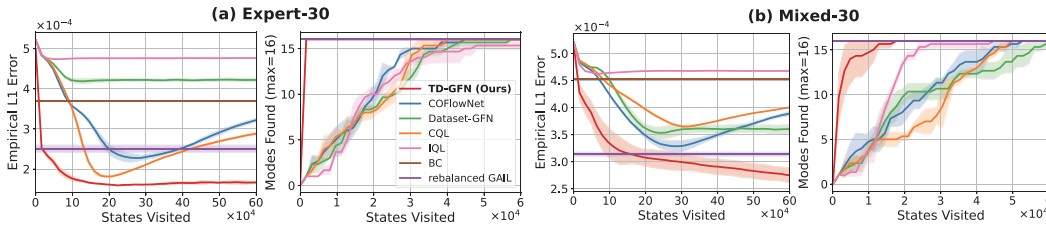

Figure 8: Performance comparison on the $8^4$ Hypergrid task using 30 trajectories from the *Expert* and *Mixed* datasets.

### D.3 ALGORITHM TRAINED ON A LARGER HYPERGRID

We evaluate the performance of TD-GFN in more challenging settings with sparser rewards, specifically using larger $20^4$ and $256^2$ Hypergrid environments. For the $20^4$ task, as shown in Figure 9, TD-GFN achieves strong performance across all four datasets, consistently and significantly outperforming COFlowNet. Furthermore, Figure 10 demonstrates that in the $256^2$ environment, TD-GFN successfully identifies all 4 widely separated modes using only 100 training trajectories. Notably, even when trained on the "Bad" dataset which contains only 3 modes, our method effectively generalizes to discover the missing mode.

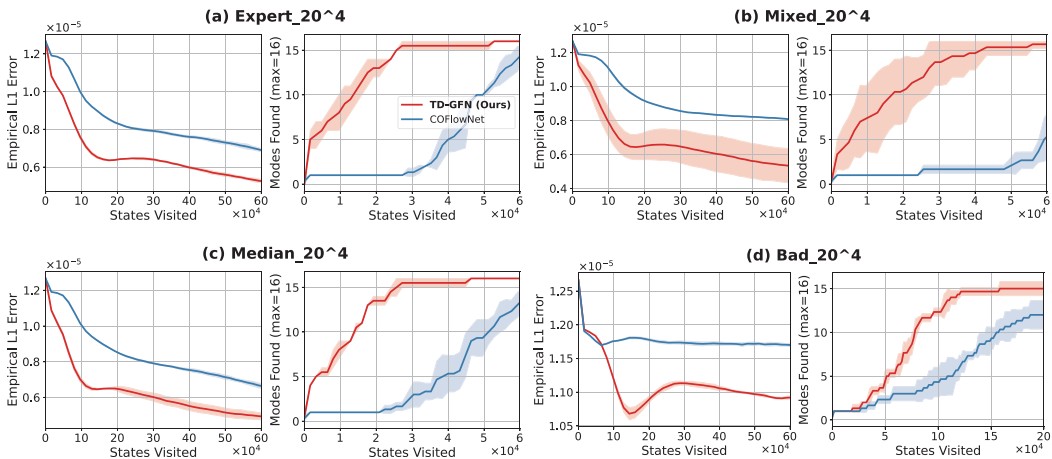

Figure 9: Performance comparison on the $20^4$ Hypergrid task using $1,500$ trajectories from four types of datasets.

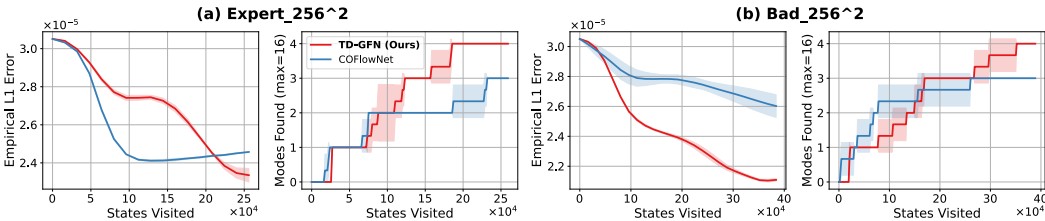

Figure 10: Performance comparison on the $256^2$ Hypergrid task using 100 trajectories from four types of datasets.

### D.4 ALGORITHM VISUALIZATION ON HYPERGRID

We visualize the behavior of out proposed TD-GFN in an $8 \times 8$ Hypergrid environment using a dataset comprising 400 transitions, constructed by combining trajectories collected from an expert policy ($50\%$) and a random policy ($50\%$). The edge rewards learned via IRL using Algorithm 2 are shown in Figure 11(b). Compared to the raw edge visitation frequencies from the dataset (Figure 11(a)), the learned edge rewards place greater emphasis on transitions that lead toward high-reward modes.

Notably, the learned edge rewards exhibit strong generalization: they assign meaningful values even to transitions that do not appear in the dataset (e.g., those in the topmost row), thereby guiding the policy toward out-of-distribution high-reward states, such as the modes in the top-right corner.

Figure 11(c) shows the pruned DAG based on the edge rewards in Figure 11(b). As illustrated, the resulting subgraph not only removes many transitions leading to low-reward regions—despite their presence in the dataset—but also constructs clear pathways toward high-reward modes, even when such paths were never encountered during data collection.

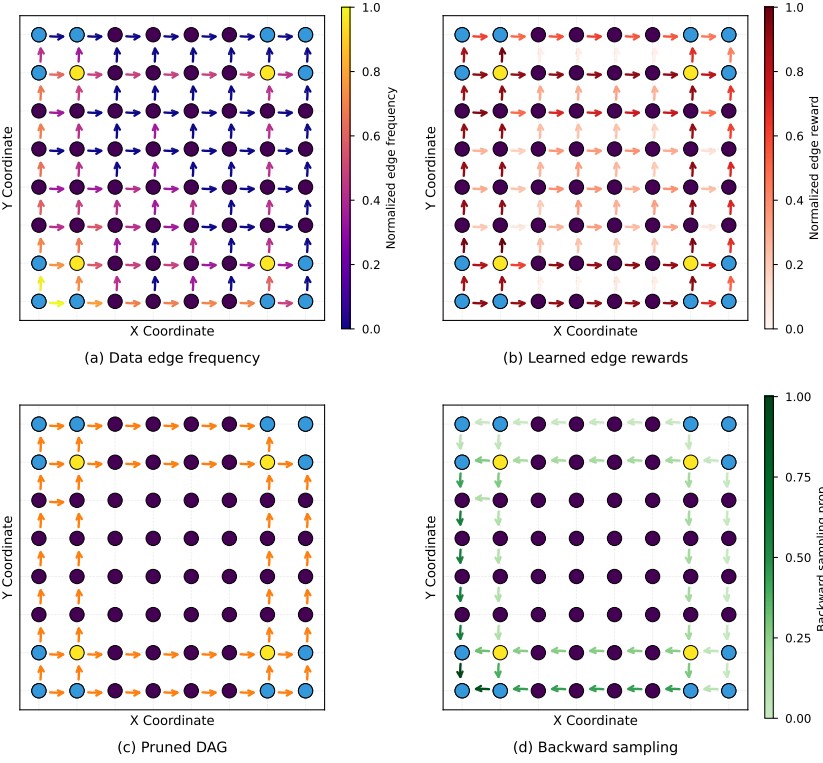

Figure 11: Visualization of TD-GFN in the $8 \times 8$ Hypergrid environment. (a) Normalized edge visitation frequencies in a dataset composed of 400 transitions obtained by combining trajectories collected separately from an expert policy ($50\%$) and a random policy ($50\%$); (b) Edge rewards learned via inverse reinforcement learning (IRL); (c) The pruned sub-DAG consisting of high-reward edges (highlighted in yellow) based on edge rewards; (d) The sampling probability of edges under backward policy induced by edge rewards and object rewards.

### D.5 SENSITIVITY ANALYSIS OF THE PRUNING THRESHOLD COEFFICIENT $K$

In our method, the pruning threshold hyperparameter $K$ is used to guide edge removal in a statistically principled manner (Equation (6)). It is tuned by selecting the value that yields the highest ratio of retained recorded transitions to retained randomly selected transitions in this work. Our experiments demonstrate that applying a fixed $K (= 7)$ across different datasets within the same environment yields strong performance without the need for dataset-specific tuning. While future work may explore more theoretically grounded strategies for DAG pruning, here we provide a sensitivity

analysis of this parameter. As shown in Figure 12, our method remains robust across different values of $K$ without catastrophic degradation.

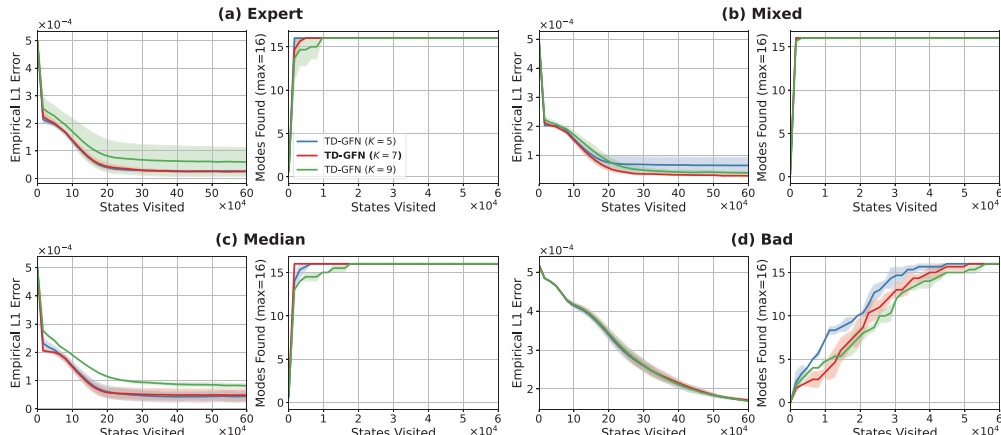

Figure 12: Performance comparison between different values of the pruning coefficient $K$ on the $8^4$ Hypergrid task. Policies are trained using four types of datasets, each consisting of $1,500$ trajectories.

### D.6 TIME CONSUMPTION AND RESOURCE USAGE

For the Molecule Design task (Section 4.3), all algorithms are trained on an NVIDIA Tesla A100 80GB GPU. We report preprocessing time, per-epoch training time, total convergence time, and peak GPU memory usage for various GFlowNet training methods in Table 5.

For Proxy-GFN, preprocessing refers to training the proxy model. For COFlowNet, it involves constructing a dictionary that indexes all child nodes from the dataset for each state. For TD-GFN, preprocessing includes both learning the edge rewards and pruning the environment DAG, corresponding to Phase 1 and Phase 2 in Algorithm 1.

Table 5: Training time and GPU memory usage comparison across GFlowNet variants on the Molecule Design task.

| Method | Preprocessing Time | Epoch Time | Convergence Time | GPU Memory |
|---|---|---|---|---|
| Proxy-GFN | 10h | 4.82s | 4h | 22.88G |
| Dataset-GFN | / | **0.56s** | 0.47h | 10.08G |
| FM-COFlowNet | 2h | 0.76s | 0.63h | 12.81G |
| QM-COFlowNet | 2h | 1.16s | 0.71h | 63.80G |
| TD-GFN (Ours) | **0.1h** | **0.59s** | **0.20h** | 12.01G |

As shown, by predicting pruned edges through a single forward pass of a neural network—rather than querying from a pre-constructed dictionary—TD-GFN significantly reduces preprocessing time and incurs minimal computational overhead during training. Furthermore, owing to its rapid convergence (as demonstrated in Table 1), TD-GFN can learn a high-performing GFlowNet policy in under 30 minutes, highlighting its strong time efficiency.

Although the edge reward learning component introduces a modest computational cost, it remains negligible compared to the overhead of more complex mechanisms such as quantile matching. Despite this, TD-GFN outperforms methods enhanced with quantile-based techniques, underscoring its effectiveness and scalability.

## D.7 ABLATION STUDY

In this section, we conduct ablation experiments on the $8^4$ Hypergrid task to evaluate the contribution of individual components within the proposed TD-GFN framework.

**Dataset Rebalancing.** As described in Section 3.1, we perform dataset rebalancing to adjust the offline data distribution to better approximate that of an expert policy. In Figure 13, we evaluate TD-GFN with and without rebalancing on both the *Expert* and *Bad* datasets. For additional comparison, we also train the offline GFlowNet method COFlowNet on the rebalanced datasets.

The results show that rebalancing the *Expert* dataset does not degrade policy learning, despite the mild distortion introduced to the original data distribution. More importantly, rebalancing significantly improves policy performance when the dataset is collected under a behavior policy that diverges substantially from the expert.

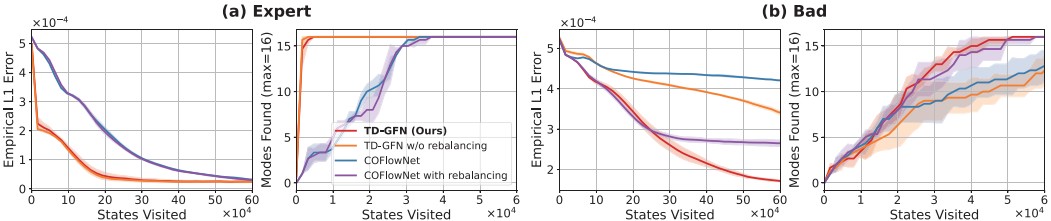

Figure 13: Performance comparison with and without dataset rebalancing on *Expert* and *Bad* datasets, each consisting of $1,500$ trajectories.

**Pruning and Backward Sampling.** To assess the individual impact of TD-GFN's components, we conduct ablations on the *Mixed* dataset. We compare the full TD-GFN with three variants: (i) a baseline without DAG pruning; (ii) a dataset-pruned variant that removes all edges not observed in the dataset; and (iii) a backward sampling baseline in which trajectories are sampled uniformly in reverse from terminal nodes, without using learned edge rewards. As shown in Figure 14, both the reward-guided pruning strategy and the prioritized backward sampling procedure are critical to the overall performance of TD-GFN.

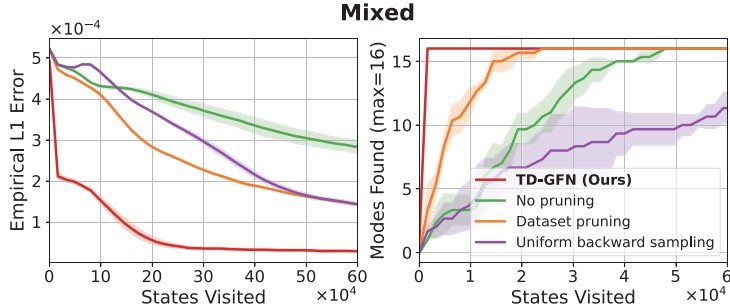

Figure 14: Ablation study on the *Mixed* dataset. We evaluate three TD-GFN variants: (i) no pruning; (ii) dataset-based pruning that removes unseen edges; and (iii) uniform backward sampling from terminal nodes. Each TD-GFN component proves essential for achieving optimal performance.

## D.8 ORTHOGONALITY TO TRAINING PARADIGMS

As shown in Algorithm 1, the improvements introduced by TD-GFN are orthogonal to the specific training paradigm employed after trajectory collection. Due to time and space constraints, our empirical analysis primarily focuses on the compatibility of TD-GFN with existing GFlowNet training objectives, as well as its integration with offline RL methods and the quantile matching technique proposed by Zhang et al. (2023b).

**Compatibility with GFlowNet training objectives.** With trajectories sampled on the pruned DAG $G' = (V', E')$, we can readily apply various GFlowNet training objectives to optimize our policy on this subgraph without requiring substantial modifications, such as the flow matching (FM) objective (Bengio et al., 2021) and the trajectory balance (TB) objective (Malkin et al., 2022), as illustrated below.

By setting $R(s) = 0$ for all non-terminal states $s$, the FM objective for each sampled trajectory $\tau$ is given by:

$$\mathcal{L}_{FM}(\tau; \theta) = \sum_{s' \in \tau} \left( \sum_{s:(s \to s') \in E'} F_\theta(s, s') - R(s') - \sum_{s'':(s' \to s'') \in E'} F_\theta(s', s'') \right)^2, \quad (8)$$

where $F_\theta(s, s')$ denotes the flow along edge $(s \to s')$. The resulting flow-based policy is then defined by sampling child states in proportion to the learned flows at each parent state.

The TB objective for each sampled trajectory $\tau$ is given by:

$$\mathcal{L}_{TB}(\tau; \theta) = \left( Z_\theta + \sum_{(s \to s') \in E'} \left( \mathcal{P}_F^{G'}(s' \mid s; \theta) - \mathcal{P}_B^{G'}(s \mid s'; \theta) \right) - R(x) \right)^2, \quad (9)$$

where $\mathcal{P}_F^{G'}(s' \mid s; \theta)$ and $\mathcal{P}_B^{G'}(s \mid s'; \theta)$ denote the forward and backward transition probabilities respectively, both normalized over the edges in the pruned subgraph $G'$ is the reward of the terminal state $x$, and $Z_\theta$ is the learned normalizing constant. Moreover, we find it beneficial to fix $\mathcal{P}_B^{G'}$ to the form given in Equation (7), a practice also noted in Malkin et al. (2022). The results of applying the TB objective on the four datasets used in our main experiments are presented in Figure 15.

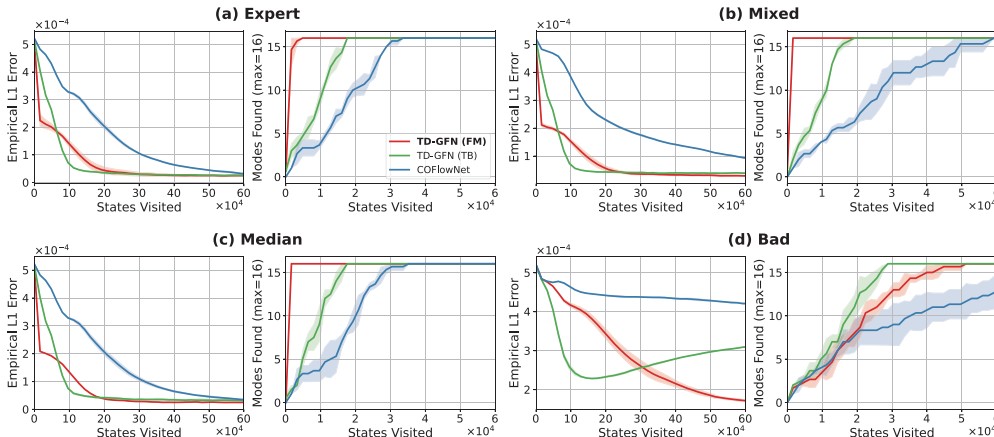

Figure 15: Performance comparison of TD-GFN with different training objectives as well as COFlowNet on the $8^4$ Hypergrid task. Policies are trained using four types of datasets, each consisting of $1,500$ trajectories.

We observe that the TB objective converges faster than the FM objective, but is less effective in rapidly exploring modes and exhibits weaker stability against overfitting on extreme datasets (e.g., *Bad*). We hypothesize this is because the flow function $F_\theta(s, s')$ in the FM objective can itself be interpreted as an implicit edge-level reward model. From this perspective, our pruning method effectively reshapes the learning landscape for this *flow-reward* model, simplifying its learning task, while our prioritized backward sampling guides it to strategically generalize to high-utility regions. Analogous to how a proxy reward model generalizes across terminal states, our approach enables this flow-reward model to generalize to unseen edges, thereby actively guiding the policy to explore novel states. This synergy might explain the FM objective's enhanced robustness and exploratory capability within our framework. Nevertheless, TD-GFN with the TB objective still substantially outperforms COFlowNet.

**Replacing GFlowNet objectives with Offline RL Training.** Instead of using the standard objectives in GFlowNet training, we examine the effect of replacing it with the Conservative Q-Learning (CQL) objective from offline reinforcement learning (Kumar et al., 2020), resulting in a variant we refer to as TD-CQL. We evaluate TD-CQL on both the *Expert* and *Mixed* datasets, each consisting of $1,500$ trajectories. Results are presented in Figure 16.

With training components inherited from TD-GFN, TD-CQL successfully discovers all modes using a considerably smaller number of state visits compared to CQL and achieves an *Empirical L1 Error* below $10^{-4}$ with fewer visits than TD-GFN, demonstrating strong sample efficiency and early-stage accuracy in modeling the target distribution. However, beyond this point, the *Empirical L1 Error* increases, indicating a decline in distributional approximation quality. This degradation arises from the nature of the offline RL objective, which tends to overfit the policy toward only the highest-reward objects, thereby limiting its ability to accurately model the full reward distribution.

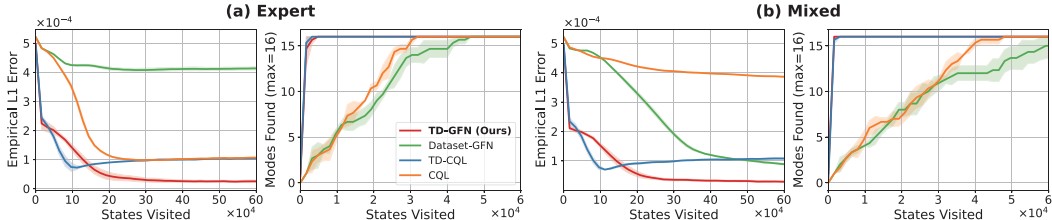

Figure 16: Performance of GFlowNet and CQL with and without our Trajectory-Distilled (TD) framework on the *Expert* and *Mixed* datasets, each consisting of $1,500$ trajectories.

**Incorporating Quantile Matching.** Following the approach of Zhang et al. (2025b), we augment TD-GFN with the quantile matching technique proposed in Zhang et al. (2023b), resulting in a variant referred to as QM-TD-GFN. We evaluate QM-TD-GFN on the Molecule Design task (Section 4.3) to assess its impact on the diversity of generated samples.

We observe that incorporating quantile matching into TD-GFN introduces significant training instability. This may be due to the pruned structure of the environment DAG, which reduces the need for a highly uncertain policy. Nevertheless, in cases where training remains stable, QM-TD-GFN is able to discover more high-reward modes than the original TD-GFN, as shown in Figure 17. These findings highlight the potential of quantile matching to enhance the diversity of generated molecules.

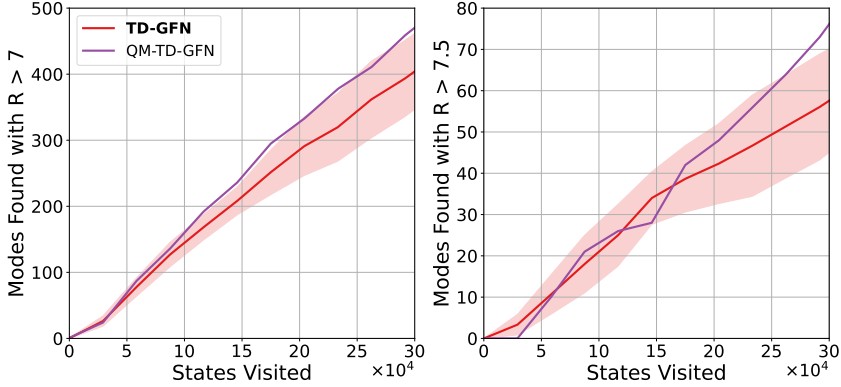

Figure 17: Number of high-reward modes discovered by QM-TD-GFN compared to TD-GFN.

### D.9 DISCUSSION ON BASELINE PERFORMANCE

To gain deeper insight into the behavior of baseline methods, we analyze results on the Hypergrid task, as shown in Figure 18.

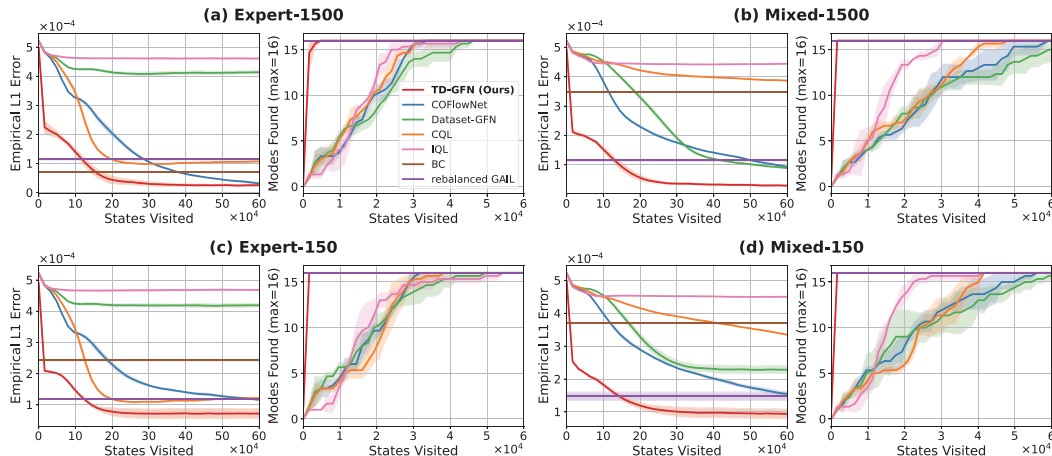

Figure 18: Performance comparison on the $8^4$ Hypergrid task. The top row uses $1,500$ trajectories from the *Expert* and *Mixed* datasets for policy training, while the bottom row uses only $150$ trajectories.

Across datasets with varying levels of uncertainty, IQL excels at identifying high-reward modes. However, it also exhibits the highest *Empirical L1 Error*, underscoring a core characteristic of offline reinforcement learning algorithms: their objective often prioritizes recovering the most rewarding trajectories over accurately modeling the full reward distribution.

Although CQL is also an offline RL algorithm, its performance differs from IQL due to a distinct training paradigm. Unlike IQL, which learns exclusively from transitions in the dataset, CQL explicitly constrains policy actions to remain close to the data by penalizing the overestimation of Q-values. As a result, its performance is highly sensitive to dataset quality. A similar sensitivity is observed in COFlowNet, which applies conservative regularization to limit its policy coverage to dataset-adjacent regions.

In contrast, GAIL—when trained on a rebalanced dataset—demonstrates stable performance across datasets. It consistently maintains coverage over all modes and achieves an *Empirical L1 Error* slightly above $1 \times 10^{-4}$. Although GAIL performs slightly worse than Behavior Cloning (BC) in scenarios with abundant expert data, it significantly outperforms BC under noisy or sparse data conditions and achieves performance comparable to COFlowNet. These results highlight the benefit of incorporating reward signals from terminal states during training by rebalancing the dataset.

Table 6: Comparison of Top-100 Reward (from Table 1) and Top-100 Internal Tanimoto Similarity (Avg. $\pm$ Std. over three seeds) on the Molecule Design task. Lower similarity indicates higher diversity.

| Method | Top-100 Internal Tanimoto Sim. ($\downarrow$) | Top-100 Reward ($\uparrow$) |
|---|---|---|
| Proxy-GFN | $0.665 \pm 0.024$ | $7.281 \pm 0.067$ |
| Oracle-GFN | $0.615 \pm 0.017$ | $\mathbf{7.408 \pm 0.021}$ |
| Dataset-GFN | $0.521 \pm 0.026$ | $7.198 \pm 0.018$ |
| FM-COFlowNet | $0.535 \pm 0.015$ | $7.201 \pm 0.015$ |
| QM-COFlowNet | $0.526 \pm 0.014$ | $7.296 \pm 0.022$ |
| **TD-GFN (Ours)** | $0.531 \pm 0.022$ | $\mathbf{7.450 \pm 0.037}$ |

## D.10 ANALYSIS OF SAMPLE DIVERSITY IN MOLECULE DESIGN

In the main paper (Figure 7), we report sample diversity using the number of high-reward modes discovered, which is a task-oriented metric evaluating the intersection of exploration and exploitation.

To provide a more direct measure of structural diversity, we performed a post-hoc analysis by computing the average internal Tanimoto similarity among the top-100 highest-reward molecules sampled from the final policies of each method. A lower Tanimoto similarity score indicates higher sample diversity.

The results are presented in Table 6. As this table shows, our algorithm does not sacrifice diversity while ensuring sampling optimality (achieving the highest Top-100 Reward). This strong balance of high reward and low internal similarity explains why, as shown in Figure 7, our method is able to discover a significantly larger number of distinct high-reward modes.

### D.11 LEARNING CURVES IN BIOSEQUENCE DESIGN TASK

We provide the learning curves in Biosequence Design task in Figure 19. We omitted them in the main paper because, while TD-GFN remains stable throughout training, baseline methods exhibit significant volatility. To ensure a fair comparison, we reported the baseline results at the checkpoints that achieve the best balance between optimality and diversity.

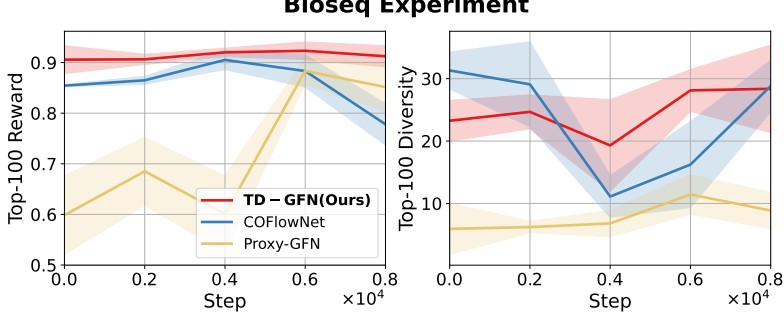

Figure 19: Learning curves of three methods in the Biosequence Design task.

### D.12 NAIVE APPLICATION OF CONVENTIONAL GFLOWNET OBJECTIVES TO OFFLINE DATASETS

To further investigate the necessity of our proposed framework, we evaluate the performance of standard GFlowNet training objectives—specifically Detailed Balance (DB) (Bengio et al., 2023), Trajectory Balance (TB) (Malkin et al., 2022) and Sub-Trajectory Balance (SubTB, with $\lambda = 0.9$) (Madan et al., 2023)—when directly applied to the offline trajectory datasets without additional guidance. Experiments were conducted on the $8^4$ Hypergrid task using both the Expert and Bad datasets ($1,500$ trajectories each).

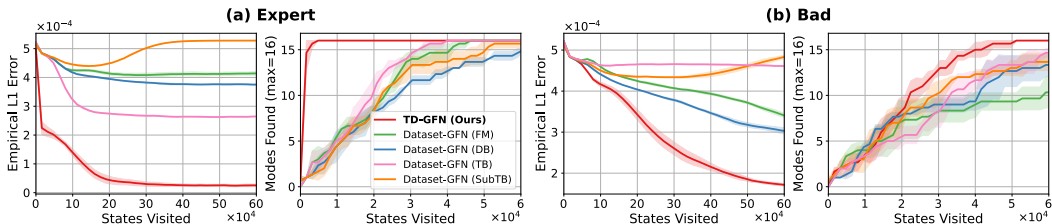

Figure 20: Performance comparison of conventional GFlowNet objectives when naively applied to $1,500$ offline trajectories in the *Expert* and *Bad* dataset on the $8^4$ Hypergrid task.

As illustrated in Figure 20, these conventional methods perform poorly compared to TD-GFN. In the absence of structural guidance for unobserved regions, simply restricting standard online objectives to fixed offline trajectories leads to suboptimal convergence and limited mode discovery. These results underscore that the naive transfer of online GFlowNet objectives to the offline setting is insufficient, highlighting the necessity of the specific mechanisms introduced in our framework.

# E DISCUSSIONS

**Why Can TD-GFN "Surpass" an Oracle-GFN?** It is remarkable that TD-GFN achieves a slightly higher average top-$k$ reward with significantly fewer training steps than an Oracle-GFN trained directly with the ground-truth reward model in the Molecule Design task. However, we emphasize that these approaches are designed for **fundamentally different settings**. We present the oracle-based results solely to benchmark the performance gap between offline training **on the current limited dataset** and ideal online training interacting with the **real environment**, rather than to claim one method is "stronger" than the other. Furthermore, while the top-$k$ reward is widely adopted as a composite metric reflecting both optimality and diversity (Bengio et al., 2021; Jain et al., 2022; Pan et al., 2022; Madan et al., 2025), we acknowledge that, as a task-oriented metric, it may carry inherent biases, as it does not directly measure the fundamental GFlowNet objective of distribution matching.

Notwithstanding these considerations, we offer two primary hypotheses to explain this outcome.

The first hypothesis relates to the exploration efficiency of the learning frameworks. A proxy-based GFlowNet, including one guided by an oracle, learns via a trial-and-error process of exploration and subsequent policy updates. Given the vastness of the search space, it is plausible that this exploration only covers a fraction of the high-reward areas that the oracle is aware of, leading the final policy to concentrate on a limited set of modes. TD-GFN, by distilling guidance from offline trajectories, may learn a more comprehensive exploration strategy that generalizes better beyond the specific samples in the dataset to a wider range of high-reward candidates.

The second hypothesis concerns the manner in which reward signals are utilized. The process of training a proxy or oracle aims to approximate the reward function over the entire space, treating different reward values with a uniform objective (e.g., minimizing mean squared error). In contrast, TD-GFN's methodology is more aligned with backward planning, which naturally prioritizes valuable goals. By rebalancing the dataset and using prioritized backward sampling, our framework explicitly emphasizes high-reward terminal states. The resulting edge rewards learned through IRL are not a simple reward surrogate but a form of dense, structural guidance, potentially highlighting more efficient paths to top-tier candidates within that chemical space.

**Adaptability to Extended GFlowNet Settings.** TD-GFN can be directly applied to scenarios involving non-acyclic directed graphs (Brunswic et al., 2024), as it operates over general transition structures without a fundamental reliance on acyclicity. In contrast, adapting our framework to continuous environments would necessitate additional modeling considerations (e.g., for defining and pruning over continuous edge sets). Furthermore, extending the method to handle stochastic rewards or non-deterministic transitions (Pan et al., 2023b) represents an important direction, particularly for real-world applications where uncertainty is inherent in both reward evaluation and environmental dynamics.

**Toward More Advanced Edge Reward Modeling.** The current implementation of TD-GFN employs an adversarial IRL approach for edge reward estimation. This choice, while effective, inherits the well-known challenges of adversarial training, such as potential instability and sensitivity to hyperparameters. Therefore, the development of more stable IRL techniques presents a promising avenue for further enhancing the reliability and scalability of the TD-GFN framework. This may be achieved by leveraging recent advances in inverse reinforcement learning, such as stability-enhancing regularization (Fu et al., 2018; Kostrikov et al., 2020), representation learning–augmented IRL (Chandak et al., 2019), and score-based or energy-based IRL methods (Liu et al., 2020).

**On Dataset Generation.** Many of the datasets in our work are generated using GFlowNets. This strategy is also common in the offline reinforcement learning (RL) community (Fu et al., 2020; Gulcehre et al., 2020), where datasets are typically collected by RL policies of varying quality. However, our method is not tied to any particular data generation procedure. For instance, in Section 4.2, we employ a dataset curated from the DBAASP database (Pirtskhalava et al., 2021). We adopt GFlowNet-generated datasets not because our approach requires them, but because they provide a controlled and reproducible way to simulate realistic scenarios in which data arises from behavior policies of different optimality levels. This allows for a systematic investigation of how dataset quality influences performance. Moreover, certain real-world datasets were not included simply because no proxy-based learning baselines are available for comparison on them. In future

work, we aim to further validate the effectiveness of our approach on datasets that more closely resemble practical deployment environments.

**What is Encoded in Trajectory Data.** Our work is situated in the offline GFlowNet setting, where training is conducted on a fixed collection of reward-labeled trajectories. These trajectories—whether collected from humans, heuristics, or simulators—are often overlooked in proxy-based methods. In contrast, our approach is designed to leverage such information without resorting to proxy reward models.

In this setting, however, our method differs from proxy-based approaches in that it explicitly leverages the trajectories themselves and assumes that they encode valuable information beyond terminal rewards. A natural question, then, is what constitutes a *valuable* trajectory. To deepen our understanding of the role of trajectories in offline GFlowNet training, we construct two additional datasets from the terminal states used in the main experiments by generating trajectories in reverse: (i) Uniform, where at each step a parent node is selected uniformly at random; and (ii) Rule-based, where the parent is chosen by preferentially reducing the dimension with the smallest current value, thereby producing trajectories that avoid the central regions of the hypergrid with low rewards. We then apply TD-GFN to these two datasets and compare the results with those obtained using the original GFlowNet-generated dataset. The results are presented in Figure 21.

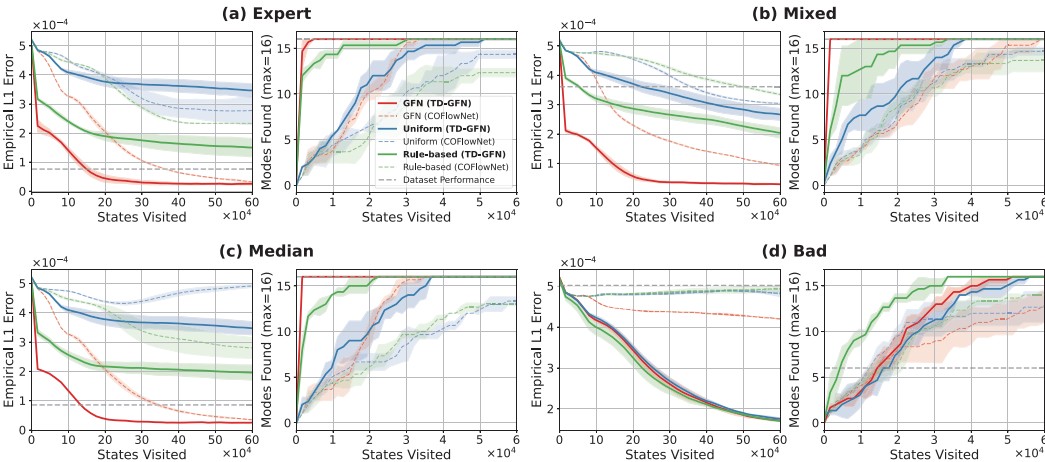

Figure 21: Performance comparison of TD-GFN policies trained on the $8^4$ Hypergrid task using three types of datasets with the same terminal states, each consisting of $1,500$ trajectories.

Although all three datasets share the same set of terminal states, the results clearly show that the trajectories themselves provide crucial information. Trajectories generated via uniform backward sampling perform very poorly, and even the rule-based trajectories, which are highly efficient and deliberately avoid low-reward regions, are less effective than trajectories generated forward by a GFlowNet. This observation suggests TD-GFN leverages meaningful trajectory structure, implying it may struggle with datasets consisting only of noise. Nevertheless, its sensitivity is mild compared to COFlowNet, which degrades more significantly under such conditions.

**Future Directions** The observation above opens up two promising directions for future work. First, developing principled methods for generating effective trajectories from terminal states could extend TD-GFN's applicability to settings where only terminal data is available.

Second, the strong performance of TD-GFN on GFlowNet-generated datasets strongly suggests its potential in online and interactive training settings. We envision that the edge rewards could be learned and updated dynamically as the agent collects new experience. This dynamic guidance model could then be used to continuously improve sampling efficiency for continued training, for instance, by intelligently focusing exploration on promising, high-reward regions of the DAG while steering the agent away from well-understood or low-utility pathways.

Furthermore, an interesting future direction would be to replace the "hard" pruning of the DAG with a "softer" guidance mechanism. In fact, the pruning method can be formally understood as an extreme case of a "soft" guidance mechanism, such as re-weighting the policy logits using the learned edge rewards. In this view, pruning is equivalent to applying an infinitely large positive weight to transitions above the threshold and a zero weight (or infinitely negative logit) to those below it. This perspective provides a clear theoretical motivation, as this re-weighting can be interpreted as imposing a strong Bayesian prior on the policy (incorporating the structural knowledge learned via IRL) or as a form of imitation learning regularization that constrains the policy to align with the distilled expert behaviors.

However, based on our initial explorations, while this "soft" method is a compelling alternative, it seems to be highly sensitive to its weighting-strength hyperparameter. A potential explanation is that it relies on the precise numerical accuracy of the learned edge rewards, which may contain generalization errors. Developing a more effective "soft" guidance mechanism remains a promising avenue for future research.

## F    STATEMENT ON LLM USAGE

In the spirit of transparency and in accordance with conference policy, we report our use of Large Language Models (LLMs) as assistive tools.

We used LLM-based tools (e.g., GPT-4o and Gemini 2.5 pro) to polish the writing in this manuscript by improving grammar, clarity, and style. For software implementation, we used the AI-powered editor Cursor to assist with routine coding tasks, such as generating boilerplate code and auto-completion.

In all instances, the LLMs served as productivity tools. The core research ideas, experimental design, model architecture, algorithmic logic, and interpretation of results are solely the work of the human authors, who bear full responsibility for all content in this paper.

