# OpenReview forum: "Beyond the Proxy: Trajectory-Distilled Guidance for Offline GFlowNet Training"
_ICLR.cc/2026/Conference — Submitted to ICLR 2026_

### Official Review · Reviewer_i231 · 2025-10-26

**Soundness:** 3
**Presentation:** 3
**Contribution:** 3
**Rating:** 6
**Confidence:** 4

**Summary:**

The paper proposes Trajectory-Distilled GFlowNet (TD-GFN), a new proxy-free GFlowNet training that eliminates the need for learned reward proxies. TD-GFN learns dense transition-level edge rewards from offline trajectories using reward-weighted inverse reinforcement learning, capturing structural importance and correlation with the rewards in the DAG. These edge rewards are used indirectly through DAG pruning and prioritized backward sampling, allowing exploration efficiency without error propagation from proxy models. Experimental results show that TD-GFN beats existing offline GFlowNet-based methods.

**Strengths:**

- The paper is well-written and easy to understand, with a clear structure and intuitive exposition.
- The motivation and proposed method are intuitive and straightforward, and the adaptation of edge-reward learning through an IRL perspective (bridging GFlowNet and RL formulations) is particularly interesting.
- The proposed method shows strong empirical results across diverse benchmarks, including Hypergrid, Biosequence, and Molecule design tasks.

**Weaknesses:**

- It is unclear why a discriminator-based formulation is necessary. Could a reward-weighted likelihood maximization over observed trajectories achieve a similar effect?
- It would be informative to analyze how well the model generates novel states unobserved during training.
- Although the method is developed for the offline setting, a discussion on how the proposed backward policy mechanism could extend to online learning would make the work more compelling.

**Questions:**

See weaknesses.

---

> ### Author Response · Authors · 2025-11-20
> **Response to Reviewer i231 (1/2)**
>
> Thank you for your time and effort in reviewing our paper!  We are very grateful for your positive assessment and for highlighting the intuitive nature of our method, particularly the **interesting adaptation** of edge-reward learning via an IRL perspective and the **strong empirical results** across diverse benchmarks.
>
> We are confident that the clarifications provided below will fully address the concerns you raised and increase your confidence in our contribution！Please note that all revisions in the updated manuscript are highlighted in **blue** for your convenience.
>
> ---
> > Weakness 1: It is unclear why a discriminator-based formulation is necessary. Could a reward-weighted likelihood maximization over observed trajectories achieve a similar effect?
>
> **Response:**
>
> Thank you for this insightful question, which touches upon the fundamental choice between imitation learning (IL) and inverse reinforcement learning (IRL) for this problem.
>
> To summarize our reasoning: we chose the discriminator-based (IRL) formulation primarily for **its superior generalization to out-of-distribution (OOD) regions**, which is critical for GFlowNets. Furthermore, we **empirically validated this choice** by including the a reward-weighted IL approach as a baseline in our experiments.
>
> **To elaborate on the first point:** you are correct that a "reward-weighted likelihood maximization" IL approach can be effective at mimicking observed expert trajectories. However, the adversarial IRL framework (specifically, the discriminator-based paradigm we adapt) learns a reward function that has been shown to provide strong generalization, enabling the policy to effectively explore and evaluate OOD regions [1, 2]. This is critical for GFlowNets, whose goal is to discover diverse, high-reward candidates, not just to reproduce those in the dataset. In fact, this paradigm has also been applied in many recent cutting-edge fields [3].
>
> > [1] Tian Xu, Ziniu Li, Yang Yu. "Error Bounds of Imitating Policies and Environments for Reinforcement Learning." NeurIPS 2020.
> >
> > [2] Fan-Ming Luo, Tian Xu, Xingchen Cao, Yang Yu. "Reward-Consistent Dynamics Models are Strongly Generalizable for Offline Reinforcement Learning." ICLR 2024.
> >
> > [3] Tianzhu Ye, Li Dong, Zewen Chi, Xun Wu, Shaohan Huang, Furu Wei. "Black-Box On-Policy Distillation of Large Language Models." arXiv:2511.10643.
>
> **To elaborate on the second point:** our "rebalanced GAIL" baseline is, in fact, **an implementation of the reward-weighted IL method**. We empirically validated this point in our experiments. As shown in our results (e.g., Figure 5, Figure 8), this baseline performs reasonably well by imitating the reward-weighted behaviors in the dataset. However, its performance degrades significantly when the dataset quality deviates from expert behavior (e.g., the "Bad" dataset) or is sparse (e.g., "Expert-30"). This performance drop highlights the exact limitation in generalization that our TD-GFN framework is designed to overcome.
>
> Therefore, the discriminator-based formulation is necessary because it provides a **generalizable** "guidance" signal (the edge rewards) rather than a "mimicry" objective, which is more robust to imperfect datasets and essential for the GFlowNet's objective of exploration and discovery. We have revised **Section 3.1** to further emphasize the critical role of this generalization capability as follows:
> > As shown in Figure 2(b) and Appendix D.4, the learned edge rewards exhibit strong generalization, assigning meaningful values even to transitions not observed in the dataset. This guides the policy to allocate attention across the entire environment DAG, rather than staying confined to regions near the training data, as in prior methods. Crucially, this generalization capability is vital for the GFlowNet objective, as covering the target distribution inherently requires the policy to navigate and evaluate transitions in regions not explicitly covered by the offline dataset.

---

> > ### Author Response · Authors · 2025-11-20
> > **Response to Reviewer i231 (2/2)**
> >
> > ---
> > > Weakness 2: It would be informative to analyze how well the model generates novel states unobserved during training.
> >
> > **Response:**
> >
> > Thank you for this valuable suggestion. In the Biosequence and Molecule Design tasks, due to the immense state space, the **vast majority** of candidates sampled by GFlowNet policies are, by definition, novel states not present in the dataset. For the Hypergrid task, we provided dataset information in Appendix D.1 of our initial submission to illustrate how many discovered modes are novel (unobserved in the dataset):
> > > "Specifically, the sampled 150 trajectories from the *Mixed* dataset and 1,500 trajectories from the *Bad* dataset cover only **12** and **6** out of 16 reward modes, respectively."
> >
> > We have moved this analysis to the main text of **Section 4.1** in the revised paper to provide a better view on how well the model generates novel states unobserved during training, as follows:
> > > Notably, the sampled $150$ trajectories from the *Mixed* dataset and $1,500$ trajectories from the *Bad* dataset cover only $\mathbf{12}$ and $\mathbf{6}$ out of $16$ reward modes, respectively. This confirms that our experiments effectively evaluate the method's capacity to generate novel states unobserved in the dataset, rather than merely assessing its degree of overfitting.
> >
> > ---
> > > Weakness 3: Although the method is developed for the offline setting, a discussion on how the proposed backward policy mechanism could extend to online learning would make the work more compelling.
> >
> > **Response:**
> >
> > Thank you for this excellent suggestion. This is precisely a direction we are actively exploring.
> >
> > As we noted in our 'Future Directions' discussion in Appendix E of our initial submission, the strong performance of TD-GFN on GFlowNet-generated datasets suggests its potential in online training. We envision that the agent's own experience could be used to maintain a dynamic edge reward model. This model could then guide sampling during both training and evaluation, and this guidance would not be limited to the "hard" DAG pruning used in our current framework. We believe this approach could fundamentally accelerate the efficiency of GFlowNet algorithms.
> >
> > We appreciate this feedback and have emphasized this point more clearly in the revised paper as follows:
> >
> > **In Appendix E "future directions" paragragh:**
> > > Second, the strong performance of TD-GFN on GFlowNet-generated datasets strongly suggests its potential in online and interactive training settings. We envision that the edge rewards could be learned and updated dynamically as the agent collects new experience. This dynamic guidance model could then be used to continuously improve sampling efficiency for continued training, for instance, by intelligently focusing exploration on promising, high-reward regions of the DAG while steering the agent away from well-understood or low-utility pathways.
> >
> > ---
> > We hope our response addresses your concerns. If so, we wonder if you could kindly consider raising your score? We will also be happy to answer any further questions you may have. Thank you very much!

---

### Official Review · Reviewer_DWcy · 2025-10-30

**Soundness:** 2
**Presentation:** 3
**Contribution:** 3
**Rating:** 4
**Confidence:** 4

**Summary:**

The paper studies the setting of learning GFlowNet from trajectory-level datasets without relying on proxy reward models. TD-GFN algorithm is proposed, which employs IRL to learn edge-level rewards from the trajectory data, after that using them to prune the environment graph and define a backward policy used for prioritized sampling to collect the data that will be utilized for training of the final GFlowNet policy. The approach is evaluated and compared to a number of baselines on hypergrid, biosequence design and molecule design environments that are standard for GFlowNet literature.

**Strengths:**

The authors study an interesting setting of learning GFlowNets from online data without relying on proxy reward models, which has high practical potential in my opinion. The paper is generally well-written and has a good structure. The presented experimental evaluation is thorough, presents a large number of baselines for comparison, studies the performance of the proposed method on data of different quality, provides various ablation studies, and uses a broad range of metrics for comparison of the algorithms, including training convergence speed, distribution approximation quality, diversity and rewards of the generated samples. The results of the experimental evaluation are promising, showing strong performance of TD-GFN on various tasks in comparison to the baselines.

**Weaknesses:**

I believe that this is a solid paper, however, there is one crucial weakness that prevents me from recommending acceptance in its current form. The setting studied in the paper is learning GFlowNets from trajectory-level data, i.e. data consisting of trajectories $s_0 \to s_1 \to \dots \to s_n$ and rewards $R(s_n)$ given for terminal states. However, the paper has no experiments on non-synthetic trajectory-level data, thus I believe it is hard to make conclusions about the potential utility of the proposed algorithm in real-world problems. Experiments on hypergrids and molecules use trajectory level datasets collected by another GFlowNet pre-trained on proxy rewards, and experiments on biosequence design use a dataset containing only terminal states $s_n$ and rewards $R(s_n)$. The latter type of data is generally more widely available and easier to collect than whole trajectories, and is the type of data used to train proxy-reward models for GFlowNets, which the paper aims at bypassing. Thus, I believe that the presented experiments do not fully fairly follow the setting that the authors claim to be studying. If the authors demonstrate the utility of the proposed approach and compare to baselines on some real-world trajectory-level dataset during the rebuttal (containing the full set of intermediate states and transitions from the initial state to the terminal state $s_0 \to s_1 \to \dots \to s_n$), I will be glad to raise my score.

In addition, even though the authors present some intuition for the proposed pruning algorithm, the paper lacks any real theoretical explanation or motivation on why it might be beneficial.

**Questions:**

1) Can you please elaborate on the proposed reward-guided pruning procedure. It may be possible that a certain edge $s \to s'$ has a low learned reward, but further edges that can be reached from it may have high rewards (since IRL works here with the setting where the policy has to maximize the sum of rewards along the trajectory plus its entropy). Still, $s \to s'$ will be pruned, potentially blocking acess to a region where the GFlowNet reward $R$ has high values. Wouldn't this be a problem? In addition, if I understand correctly, the pruning may lead to completely removing access to some parts of the environment if all edges crossing some cut in the graph are removed. This may also lead to some modes of the reward distribution remaining hidden from the GFlowNet policy.

---

> ### Author Response · Authors · 2025-11-20
> **Response to Reviewer DWcy (1/4)**
>
> Thank you for your time and effort in reviewing our paper! We are highly encouraged by your recognition of the **high practical potential** of our proxy-free setting and the **thoroughness of our experimental evaluation**.
>
> We are confident that the clarifications and additional results provided below will address the concerns you raised and increase your confidence in our contribution. Please note that all revisions in the updated manuscript are highlighted in **blue** for your convenience.
>
> ---
> > Weakness 1: I believe that this is a solid paper, however, there is one crucial weakness that prevents me from recommending acceptance in its current form. The setting studied in the paper is learning GFlowNets from trajectory-level data, i.e. data consisting of trajectories
> $s_0 \rightarrow s_1 \rightarrow \cdots \rightarrow s_n$ and rewards $R(s_n)$ given for terminal states. However, the paper has no experiments on non-synthetic trajectory-level data, thus I believe it is hard to make conclusions about the potential utility of the proposed algorithm in real-world problems. Experiments on hypergrids and molecules use trajectory level datasets collected by another GFlowNet pre-trained on proxy rewards, and experiments on biosequence design use a dataset containing only terminal states $s_n$ and rewards $R(s_n)$. The latter type of data is generally more widely available and easier to collect than whole trajectories, and is the type of data used to train proxy-reward models for GFlowNets, which the paper aims at bypassing. Thus, I believe that the presented experiments do not fully fairly follow the setting that the authors claim to be studying. If the authors demonstrate the utility of the proposed approach and compare to baselines on some real-world trajectory-level dataset during the rebuttal (containing the full set of intermediate states and transitions from the initial state to the terminal state $s_0 \rightarrow s_1 \rightarrow \cdots \rightarrow s_n$ and rewards $R(s_n)$), I will be glad to raise my score.
>
> **Response:**
>
> We sincerely appreciate your insightful comments and constructive critique regarding the nature of the datasets used in our initial submission.
>
> **Clarification on Tree-Structured Domains**
>
> First, we would like to highlight that while GFlowNets were originally proposed for general DAGs, they have demonstrated exceptional potential in environments modeled as trees, such as Biosequence Design and LLM fine-tuning [1,2].
>
> > [1] Fangxu Yu, Lai Jiang, Haoqiang Kang, Shibo Hao, Lianhui Qin. "Flow of Reasoning: Training LLMs for Divergent Reasoning with Minimal Examples." ICML 2025.
> >
> > [2] Xuekai Zhu, Daixuan Cheng, Dinghuai Zhang, Hengli Li, Kaiyan Zhang, Che Jiang, Youbang Sun, Ermo Hua, Yuxin Zuo, Xingtai Lv, Qizheng Zhang, Lin Chen, Fanghao Shao, Bo Xue, Yunchong Song, Zhenjie Yang, Ganqu Cui, Ning Ding, Jianfeng Gao, Xiaodong Liu, Bowen Zhou, Hongyuan Mei, Zhouhan Lin. "FlowRL: Matching Reward Distributions for LLM Reasoning." arXiv:2509.15207.
>
> In such environments, while widely available datasets often contain only terminal states, the generation trajectories can be **uniquely inferred** from the final output. We emphasize that our TD-GFN is **fully applicable** to these scenarios and effectively exploits the inferred trajectory structure.
>
> **New Experiment on Real-World Trajectories**
>
> To directly address your concern and demonstrate the utility of our approach on **real-world, non-synthetic trajectory-level data**, we have incorporated an additional experiment on the Listwise Recommendation task using the MovieLens 1M (ML1M) dataset [3], a standard benchmark containing real user-item interactions.
>
> > [3] Shuchang Liu, Qingpeng Cai, Zhankui He, Bowen Sun, Julian McAuley, Dong Zheng, Peng Jiang, Kun Gai. "Generative Flow Network for Listwise Recommendation." ACM KDD 2023.
>
> * **Task Definition:** The agent is tasked with autoregressively generating a list (slate) of $K=6$ unique items from a large discrete action space of 3,706 items.
> * **Real-World Trajectories:** We utilized a subset of the dataset containing **real user interaction trajectories** to simulate the offline setting. Intermediate states consist of the list of movies already watched and rated by the user. A reward is assigned once $K=6$ items are rated, calculated as the average of the user's ratings.
> * **Non-Degenerate DAG Structure:** Crucially, since the quality of a recommendation list is invariant to the relative order of items, this environment constitutes a non-degenerate DAG rather than a tree, as different generation orders lead to the same terminal set.

---

> ### Author Response · Authors · 2025-11-20
> **Response to Reviewer DWcy (2/4)**
>
> [Continuation of Response to Weakness 1]
> The results are presented in the table below:
>
> | Method | Avg. Reward ($\uparrow$) | Max Reward ($\uparrow$) | Coverage ($\uparrow$) |
> | :--- | :---: | :---: | :---: |
> | Oracle-GFN | **2.092 $\pm$ 0.033** | **2.930 $\pm$ 0.017** | 113.7 $\pm$ 72.0 |
> | Dataset-GFN | 1.681 $\pm$ 0.202 | 2.674 $\pm$ 0.184 | 7.9 $\pm$ 0.9 |
> | QM-COFlowNet | 1.787 $\pm$ 0.022 | 2.843 $\pm$ 0.020 | 119.2 $\pm$ 6.1 |
> | **TD-GFN (Ours)** | **1.988 $\pm$ 0.006** | **2.914 $\pm$ 0.013** | **186.8 $\pm$ 6.3** |
>
> As shown, TD-GFN consistently outperforms other offline GFN baselines (Dataset-GFN and QM-COFlowNet). Notably, it achieves reward performance competitive with the Oracle-GFN, which learning directly from the oracle environment, while exhibiting significantly higher and more stable diversity (Coverage).
>
> We have included these new experimental results and detailed settings in **Appendix D.1** of the revised paper. We envision that our approach can potentially unlock the potential for GFlowNet applications across a broader range of domains and scenarios.

---

> ### Author Response · Authors · 2025-11-20
> **Response to Reviewer DWcy (3/4)**
>
> ---
> > Weakness 2: In addition, even though the authors present some intuition for the proposed pruning algorithm, the paper lacks any real theoretical explanation or motivation on why it might be beneficial.
>
> **Response:**
>
> Thank you for this valuable suggestion. Though we did not include a formal theoretical analysis in this work, we can clarify the theoretical motivation here.
>
> Prior work has identified structural challenges in GFlowNet training: [4] reveals how unobserved transitions lead GFlowNets to under-exploit high-reward objects, while [5] demonstrates that imbalances across edges can systematically compromise sampling accuracy. Both studies address these issues by employing reweighting strategies to **emphasize edges deemed beneficial for the GFlowNet objective**.
>
> > [4] Hyosoon Jang, Yunhui Jang, Minsu Kim, Jinkyoo Park, Sungsoo Ahn. "Pessimistic Backward Policy for GFlowNets." NeurIPS 2024.
> >
> > [5] Tiago Silva, Rodrigo Barreto Alves, Eliezer de Souza da Silva, Amauri H Souza, Vikas Garg, Samuel Kaski, Diego Mesquita. "When do GFlowNets learn the right distribution?" ICLR 2025.
>
> In this context, our 'hard' pruning method can be formally understood as an **extreme case of a 'soft' guidance mechanism**, such as re-weighting the policy logits using the learned edge rewards. In this view, pruning is equivalent to applying an infinitely large positive weight to transitions above the threshold and a zero weight (or infinitely negative logit) to those below it.
>
> This perspective provides a clear theoretical motivation, as this re-weighting can be interpreted as imposing a strong **Bayesian prior** on the policy (incorporating the structural knowledge learned via IRL) or as a form of **imitation learning regularization** that constrains the policy to align with the distilled "expert" behaviors.
>
> Moreover, our proposed mechanism exhibits a particularly strong synergy with the Flow Matching (FM) objective. Under FM, the flow function $F_\theta(s, s')$ effectively functions as **an implicit edge-level reward model**. From this perspective, our pruning method reshapes the learning landscape for this ''flow-reward'' model, simplifying its optimization task, while our prioritized backward sampling guides it to generalize strategically toward high-utility regions. Analogous to how a proxy reward model generalizes across terminal states, our approach enables this flow-reward model to generalize to unseen edges, thereby actively guiding the policy to explore novel states.
>
> We agree that a detailed, formal analysis of these connections is a very promising direction for future work. We have added this discussion to our revised paper to better motivate our design as follows:
>
> **In Appendix D.8**
> > We hypothesize this is because the flow function $F_\theta(s, s')$ in the FM objective can itself be interpreted as an implicit edge-level reward model. From this perspective, our pruning method effectively reshapes the learning landscape for this 'flow-reward' model, simplifying its learning task, while our prioritized backward sampling guides it to strategically generalize to high-reward regions. Analogous to how a proxy reward model generalizes across terminal states, our approach enables this flow-reward model to generalize to unseen edges, thereby actively guiding the policy to explore novel states. This synergy might explain the FM objective's enhanced robustness and exploratory capability within our framework.
>
> **In Appendix E**
> > Furthermore, an interesting future direction would be to replace the “hard” pruning of the DAG with a “softer” guidance mechanism. In fact, the pruning method can be formally understood as an extreme case of a “soft” guidance mechanism, such as re-weighting the policy logits using the learned edge rewards. In this view, pruning is equivalent to applying an infinitely large positive weight to transitions above the threshold and a zero weight (or infinitely negative logit) to those below it. This perspective provides a clear theoretical motivation, as this re-weighting can be interpreted as imposing a strong Bayesian prior on the policy (incorporating the structural knowledge learned via IRL) or as a form of imitation learning regularization that constrains the policy to align with the distilled expert behaviors.

---

> > ### Author Response · Authors · 2025-11-20
> > **Response to Reviewer DWcy (4/4)**
> >
> > > Question 1: Can you please elaborate on the proposed reward-guided pruning procedure. It may be possible that a certain edge $s \rightarrow s'$ has a low learned reward, but further edges that can be reached from it may have high rewards (since IRL works here with the setting where the policy has to maximize the sum of rewards along the trajectory plus its entropy). Still, $s \rightarrow s'$ will be pruned, potentially blocking acess to a region where the GFlowNet reward $R$ has high values. Wouldn't this be a problem? In addition, if I understand correctly, the pruning may lead to completely removing access to some parts of the environment if all edges crossing some cut in the graph are removed. This may also lead to some modes of the reward distribution remaining hidden from the GFlowNet policy.
> >
> > **Response:**
> >
> > Thank you for this very deep and insightful question about the pruning procedure. It targets the core of our method's design. You are right that an edge $s \rightarrow s'$ with a low learned edge reward might exist on a path to a terminal state with a high GFlowNet reward $R(x)$. However, this is actually **consistent with the intended behavior of our method**.
> >
> > Based on the theoretical equivalence between GFlowNet training and entropy-regularized RL (Equations (1) and (2) in our paper), the learned edge reward (ignoring learning error) represents the utility of that edge for learning an optimal GFlowNet policy. The GFlowNet objective is not just to find high-reward states, but to match the target distribution $R(x)$, which implicitly involves maximizing causal entropy.
> >
> > Therefore, the scenario you described—a low edge reward on a path to a high terminal reward—correctly identifies a path with very low causal entropy. Forcing the GFlowNet to use this path would make it prone to collapsing into a local optimum (like a simple RL agent) rather than optimizing its true objective of distribution matching. Pruning this edge is the *correct* action, as it guides the policy away from "narrow" exploitative paths that would harm the final distribution.
> >
> > In the realistic scenario where the IRL model inevitably incurs generalization errors, our pruning mechanism does not require perfect precision. It suffices for the edge reward model to be **sufficient at the task of distinguishing low-utility edges from potentially useful ones**, thereby mitigating the impact of numerical learning errors. While it is theoretically possible that extreme learning error in the IRL phase could lead to mistakenly pruning paths, it is unrealistic to expect the policy to generalize effectively on transitions where the learned edge rewards exhibit such significant deviations.
> >
> > Empirically, our extensive results demonstrate that TD-GFN remains highly effective despite these potential imperfections. Our analysis in **Appendix C.4 and D.5** shows the method is **robust to the pruning threshold hyperparameter**. Furthermore, our main experiments in **Section 4.1 and Appendix D.2** show that TD-GFN consistently discovers all modes and outperforms baselines, even on highly noisy ('Mixed', 'Bad') and extremely sparse ('Expert-150', 'Expert-30') datasets. This strongly suggests that our pruning method effectively guides the policy to find modes, rather than incorrectly hiding them.
> >
> > We have elaborated on the rationale for adopting this pruning strategy in **Section 3.2** of the revised paper as follows:
> > > By adopting this threshold-based pruning strategy, we rely primarily on the model’s ability to distinguish low-reward edges from potentially useful ones, thereby mitigating the risk of policy degradation caused by potential numerical errors in the learned IRL rewards.
> >
> > ---
> > We hope our response addresses your concerns. If so, we wonder if you could kindly consider raising your score? We will also be happy to answer any further questions you may have. Thank you very much!

---

### Official Review · Reviewer_cpmu · 2025-10-31

**Soundness:** 2
**Presentation:** 3
**Contribution:** 2
**Rating:** 2
**Confidence:** 5

**Summary:**

The paper proposes Trajectory-Distilled GFlowNet (TD-GFN), a proxy-free method for training GFlowNet based on rewards from pre-collected offline trajectories rather than a learned reward function. It (1) uses GAIL on a pre-collected dataset to learn edge-level rewards over the environment DAG; (2) prunes low-utility edges using a threshold; and (3) performs prioritized backward sampling on the pruned DAG graph while updating the forward policy only with ground-truth terminal rewards from the dataset. Experiments on Hypergrid, antimicrobial peptide (AMP) design, and molecular design show faster convergence and stronger top-k rewards vs. proxy-based, proxy-free, imitation, and offline-RL baselines.

**Strengths:**

* Turning trajectory data into dense, structural guidance via IRL-derived edge rewards is new in the GFlowNet offline setting.

* The pruning criterion and prioritized backward sampler are simple and interpretable.

* Pipeline and training objectives are well explained.

**Weaknesses:**

1. The claimed advantages of TD-GFN rest primarily on HyperGrid benchmarks, but the chosen grid size ($8^4$) is too small to offer convincing evidence. Moreover, HyperGrids are homogeneous across dimensions, so grid height (also equal to the trajectory length) matters far more than dimensionality. With such a short trajectory length ($=8$), environment-DAG pruning becomes less meaningful.   Compared to girds with large trajectory length (e.g.  $256$ and $512$),  just making random transitions can easily traverse the reward landscapes and reach all reward modes.

2. It is misleading to state that proxy-free methods based on pre-collected rewards and trajectories is more efficient than learning from the reward oracle.  Since your trajectories are simulated (we can only observe the terminal states in reality) and your rewards are computed from the oracle, this can not happens.  The point is achieving higher top-k rewards does not establish a better GFlowNets as the goal is capturing distributions, which demands both optimality and diversity.

3. The claim that proxy-free training is superior to proxy-based approaches is overstated. TD-GFN is “proxy-free” in that it trains on simulated trajectories terminating in a small subset of $\mathcal{X}$ with provided rewards. Conventional offline GFlowNet methods (e.g., TB, DB, Sub-TB) could adopt the same setup by restricting training to those trajectories; in other words, these methods can be run in proxy-based, proxy-free, or hybrid modes.

4. The necessity of proxy-free training on purely offline trajectories is **not** justified for the tasks considered, for two reasons:
    * First, in domains like self-driving, proxy-free training is needed because we must not only achieve high terminal rewards but also mimic the actions/transitions along pre-collected trajectories. By contrast, in typical GFlowNet applications, only terminal-state signals are observed in the real world and the intermediate trajectories are fully simulated—so the “offline-only trajectories” argument does not apply.

    * Seconds, the meaning of learning proxies from provided data is to obtain reward values over the entire terminal space; while imperfect, such proxies can facilitate exploration beyond $\mathcal{X}$ and discover new modes. This further underscores why capturing the overall terminal distributions matters[1].

5. If the stated advantage of TD-GFN is better performance (in terms of either capturing distribution or discovering modes) than proxy-based GFNs, the evaluation does not establish this fairly. Most results compare against proxy-free GFN/RL baselines; rigorous baselines using conventional proxy-based GFlowNets (e.g., TB, Sub-TB with learned proxies) are absent or underreported. Moreover, because conventional proxy-based approaches are offline, there are substantial prior works on offline data samplers with replay buffers (e.g. [2]). As they does not touch the conventional training objectives, these GFN methods are clearly more straightforward than the proposed tree-phase pipeline ( IRL > DAG-pruning > policy updating) and should be included for a fair comparison.

6. While it is claimed by the authors,  the paper does not demonstrate a true exploration–exploitation balance. Exploration (novel terminal-state visitation) and exploitation (concentration on known high-reward terminals) are not controlled. Because TD-GFN is proxy-free, it primarily optimizes a policy around the given limited set of rewarded terminals, i.e., exploitation.

[1] Jain, Moksh, Tristan Deleu, Jason Hartford, Cheng-Hao Liu, Alex Hernandez-Garcia, and Yoshua Bengio. "Gflownets for ai-driven scientific discovery." Digital Discovery 2, no. 3 (2023): 557-577.

[2] Kim, Minsu, et al. "Local Search GFlowNets." The Twelfth International Conference on Learning Representations.

**Questions:**

* Can you provide experiment results on grids with long trajectory length? (e.g.  $128^3$,   $256^2$, and $512^2$).

* Why the training curves of AMP tasks are not provided?

* Where is the diversity reporting for sEH task?

---

> ### Author Response · Authors · 2025-11-20
> **Response to Reviewer cpmu (1/7)**
>
> Thank you for your time and effort in reviewing our paper! We are grateful for your constructive suggestions, which have significantly guided our improvements.
>
> We try our best to address your questions as follows. Please note that all revisions in the updated manuscript are highlighted in **blue** for your convenience.
>
> ---
> > Weakness 1: The claimed advantages of TD-GFN rest primarily on HyperGrid benchmarks, but the chosen grid size ($8^4$) is too small to offer convincing evidence. Moreover, HyperGrids are homogeneous across dimensions, so grid height (also equal to the trajectory length) matters far more than dimensionality. With such a short trajectory length (=8), environment-DAG pruning becomes less meaningful. Compared to girds with large trajectory length (e.g. 256 and 512), just making random transitions can easily traverse the reward landscapes and reach all reward modes.
>
> > Question 1: Can you provide experiment results on grids with long trajectory length? (e.g. $128^3$, $256^2$, and $512^2$).
>
> **Response:**
>
> Thank you for your insightful comments.
>
> First, we wish to clarify that the $8^4$ Hypergrid is a standard and challenging benchmark in the GFlowNet literature [1,2]. It is not trivially solvable by random transitions; the Manhattan distance between the two farthest reward modes requires approximately 28 steps to traverse, posing a significant exploration challenge particularly in our offline setting. Furthermore, as detailed in Appendix D.2 of our original submission, we have already demonstrated the efficacy of TD-GFN on a much larger $20^4$ Hypergrid.
>
> > [1] Emmanuel Bengio, Moksh Jain, Maksym Korablyov, Doina Precup, Yoshua Bengio. "Flow Network based Generative Models for Non-Iterative Diverse Candidate Generation." NeurIPS 2021.
> >
> > [2] Nikolay Malkin, Moksh Jain, Emmanuel Bengio, Chen Sun, Yoshua Bengio. "Trajectory balance: Improved credit assignment in GFlowNets." NeurIPS 2022.
>
>
> To fully address your concerns regarding environments with significantly longer trajectory lengths, we conducted additional experiments on a $256^2$ Hypergrid and have incorporated the results into Appendix D.3 of the revised paper.
>
> As illustrated, TD-GFN successfully identifies all 4 widely separated reward modes using only **100** training trajectories. Most notably, even when trained on the *Bad* dataset—which contains only **3** of the 4 modes—our method effectively generalizes to discover the missing mode. We believe these results strongly validate the effectiveness of our environment-DAG pruning and edge-reward guidance in long-horizon tasks.

---

> ### Author Response · Authors · 2025-11-20
> **Response to Reviewer cpmu (2/7)**
>
> ---
> > Weakness 2: It is misleading to state that proxy-free methods based on pre-collected rewards and trajectories is more efficient than learning from the reward oracle. Since your trajectories are simulated (we can only observe the terminal states in reality) and your rewards are computed from the oracle, this can not happens. The point is achieving higher top-k rewards does not establish a better GFlowNets as the goal is capturing distributions, which demands both optimality and diversity.
>
> **Response:**
>
> We thank the reviewer for this insightful comment. While we acknowledge that the **Mean Top-K Reward** is a **task-oriented** metric, it effectively serves as a composite indicator reflecting both **optimality** and **diversity**. To achieve a high score on this metric (computed on unique candidates), an algorithm is strictly required to sample a *diverse* set of *high-reward* candidates, rather than repeatedly sampling a single local optimum. Consequently, it is a comprehensive standard for policy evaluation **widely adopted** in GFlowNet literature, e.g., [3-5].
>
> > [3] Moksh Jain, Emmanuel Bengio, Alex-Hernandez Garcia, Jarrid Rector-Brooks, Bonaventure F. P. Dossou, Chanakya Ekbote, Jie Fu, Tianyu Zhang, Micheal Kilgour, Dinghuai Zhang, Lena Simine, Payel Das, Yoshua Bengio. "Biological sequence design with gflownets." ICML 2022.
> >
> > [4] Ling Pan, Dinghuai Zhang, Aaron Courville, Longbo Huang, Yoshua Bengio. "Generative Augmented Flow Networks." ICLR 2023.
> >
> > [5] Kanika Madan, Alex Lamb, Emmanuel Bengio, Glen Berseth, Yoshua Bengio. "Towards Improving Exploration through Sibling Augmented GFlowNets." ICLR 2025.
>
> To quantify diversity more explicitly, we have added a post-hoc analysis of the pairwise Tanimoto similarity among the top-100 sampled molecules in **Appendix D.10** of the revised paper (see table below). These results demonstrate that **TD-GFN achieves superior optimality without sacrificing diversity**.
>
> | Method | Top-100 Internal Tanimoto Sim. ($\downarrow$) | Top-100 Reward ($\uparrow$) |
> | :--- | :---: | :---: |
> | Proxy-GFN | 0.665 $\pm$ 0.024 | 7.281 $\pm$ 0.067 |
> | Oracle-GFN | 0.615 $\pm$ 0.017 | 7.408 $\pm$ 0.021 |
> | Dataset-GFN | 0.521 $\pm$ 0.026 | 7.198 $\pm$ 0.018 |
> | FM-COFlowNet | 0.535 $\pm$ 0.015 | 7.201 $\pm$ 0.015 |
> | QM-COFlowNet | 0.526 $\pm$ 0.014 | 7.296 $\pm$ 0.022 |
> | **TD-GFN (Ours)** | **0.531 $\pm$ 0.022** | **7.450 $\pm$ 0.037** |
>
> Regarding the comparison with oracle-based methods: given that the performance of proxy-free approaches is inherently constrained by the quality of the fixed offline dataset, we present the results of methods interacting with the oracle solely **for reference**. This comparison serves to benchmark the performance gap between offline training **on the current dataset** and ideal online training with **access to the ground-truth environment**, rather than to imply a direct competition, as their application scenarios are **fundamentally different**.
>
> We have clarified these points in our revised paper as follows:
>
> **In Introduction**
> > By leveraging these complete, pre-collected paths rather than relying on trial-and-error exploration, these methods converge to their optimal performance significantly faster during training,...
>
> > As shown in Figure 1, it achieves performance remarkably competitive with the Oracle-GFN,...
>
> **In Appendix E**
> > **Why Can TD-GFN “Surpass” an Oracle-GFN?** It is remarkable that TD-GFN achieves a slightly higher average top-k reward with significantly fewer training steps than an Oracle-GFN trained directly with the ground-truth reward model in the Molecule Design task. However, we emphasize that these approaches are designed for **fundamentally different settings**. We present the oracle-based results solely to benchmark the performance gap between offline training **on the current limited dataset** and ideal online training interacting with the **real environment**, rather than to claim one method is “stronger” than the other. Furthermore, while the top-k reward is widely adopted as a composite metric reflecting both optimality and diversity (Bengio et al., 2021; Jain et al., 2022; Pan et al., 2022; Madan et al., 2025), we acknowledge that, as a task-oriented metric, it may carry inherent biases, as it does not directly measure the fundamental GFlowNet objective of distribution matching.

---

> ### Author Response · Authors · 2025-11-20
> **Response to Reviewer cpmu (3/7)**
>
> ---
> > Weakness 3: The claim that proxy-free training is superior to proxy-based approaches is overstated. TD-GFN is “proxy-free” in that it trains on simulated trajectories terminating in a small subset of with provided rewards. Conventional offline GFlowNet methods (e.g., TB, DB, Sub-TB) could adopt the same setup by restricting training to those trajectories; in other words, these methods can be run in proxy-based, proxy-free, or hybrid modes.
>
> **Response:**
>
> Thank you for your question. We wish to clarify that we *do not* claim our proxy-free method is inherently superior to proxy-based approaches in general. Our research is primarily concerned with a specific but critical scenario: how to utilize historical generation data to train a functional GFlowNet when interaction with the real environment is impossible and the available dataset is insufficient to train a high-fidelity proxy. Essentially, our work addresses the practical question: **"How should we proceed when a reliable proxy is unavailable?"**
>
> In our experiments, the inclusion of the Proxy-GFN baseline, which learns from reward signals provided by a proxy trained on the terminal states in our dataset, serves precisely to demonstrate that under data-limited conditions, it is often unfeasible to train a proxy that aligns well with the oracle. This comparison validates the value of our approach in such data-constrained settings. We apologize for any ambiguity and sincerely appreciate your constructive feedback. Accordingly, we have explicitly clarified this motivation in our revised paper as follows:
>
> **In Abstract**
> > The prevailing training methods rely on a proxy model to provide reward feedback for online sampled trajectories. However, in scenarios where constructing a reliable proxy is challenging due to data scarcity or cost, one must turn to static offline trajectories for training. Nevertheless, current proxy-free approaches often rely on coarse constraints that may limit the model’s ability to explore.
>
> **In Introduction**
> > Conversely, execution trajectories in these domains are inherently generated and readily available. While often overlooked in proxy-based methods, this trajectory-level information is a cornerstone of offline reinforcement learning (Kumar et al., 2020; Kostrikov et al., 2021) that has yet to be fully exploited for GFlowNets. As recent works expand GFlowNets into a broader spectrum of domains (Liu et al., 2023; Hu et al., 2024a; Zhu et al., 2025) to enable diverse generation, a critical question emerges: how can we effectively leverage these offline trajectories to train GFlowNets when the available terminal rewards are insufficient to construct a reliable proxy model?
>
> > Through extensive experiments, we demonstrate that TD-GFN establishes a new state of the art for training GFlowNets directly from offline trajectories.
>
> Regarding conventional methods, you are correct that conventional objectives such as TB and Sub-TB can indeed be directly employed for training on offline trajectories. However, as demonstrated by the supplementary results in **Appendix D.12** of the revised paper, these methods **perform poorly** in the absence of guidance for unobserved regions. This observation indicates that the naive transfer of online training objectives to the offline setting is **insufficient**, thereby further underscoring the necessity and contribution of our proposed framework.

---

> ### Author Response · Authors · 2025-11-20
> **Response to Reviewer cpmu (4/7)**
>
> ---
> > Weakness 4.1: The necessity of proxy-free training on purely offline trajectories is not justified for the tasks considered, for two reasons:
> > - First, in domains like self-driving, proxy-free training is needed because we must not only achieve high terminal rewards but also mimic the actions/transitions along pre-collected trajectories. By contrast, in typical GFlowNet applications, only terminal-state signals are observed in the real world and the intermediate trajectories are fully simulated—so the “offline-only trajectories” argument does not apply.
>
> **Response:**
>
> Thank you for your insightful question. We respectfully submit that the necessity of learning from offline trajectories is well-founded, as GFlowNet applications are rapidly expanding into high-impact domains where trajectories are either uniquely implied by terminal states or naturally preserved as real-world interaction data.
>
> First, we would like to highlight that while GFlowNets were originally proposed for general DAGs, they have demonstrated exceptional potential in **environments modeled as trees**, such as Biosequence Design and LLM fine-tuning [6,7]. In such environments, while widely available datasets often contain only terminal states, the generation trajectories can be **uniquely inferred** from the final output. We emphasize that our TD-GFN is **fully applicable** to these scenarios and effectively exploits the inferred trajectory structure.
>
> > [6] Fangxu Yu, Lai Jiang, Haoqiang Kang, Shibo Hao, Lianhui Qin. "Flow of Reasoning: Training LLMs for Divergent Reasoning with Minimal Examples." ICML 2025.
> >
> > [7] Xuekai Zhu, Daixuan Cheng, Dinghuai Zhang, Hengli Li, Kaiyan Zhang, Che Jiang, Youbang Sun, Ermo Hua, Yuxin Zuo, Xingtai Lv, Qizheng Zhang, Lin Chen, Fanghao Shao, Bo Xue, Yunchong Song, Zhenjie Yang, Ganqu Cui, Ning Ding, Jianfeng Gao, Xiaodong Liu, Bowen Zhou, Hongyuan Mei, Zhouhan Lin. "FlowRL: Matching Reward Distributions for LLM Reasoning." arXiv:2509.15207.
>
> Second, with the increasing adoption of GFlowNets in emerging high-impact domains, we encounter **fields rich in real-world user interaction data (trajectories)**, such as Recommendation Systems [8,9] and personalized content generation [10]. To demonstrate this, we have incorporated an additional experiment on the **Listwise Recommendation** task using the MovieLens 1M (ML1M) dataset [8], a standard benchmark containing real user-item interactions.
>
> > [8] Shuchang Liu, Qingpeng Cai, Zhankui He, Bowen Sun, Julian McAuley, Dong Zheng, Peng Jiang, Kun Gai. "Generative Flow Network for Listwise Recommendation." ACM KDD 2023.
> >
> > [9] Yejing Wang, Shengyu Zhou, Jinyu Lu, Qidong Liu, Xinhang Li, Wenlin Zhang, Feng Li, Pengjie Wang, Jian Xu, Bo Zheng, Xiangyu Zhao. "GFlowGR: Fine-tuning Generative Recommendation Frameworks with Generative Flow Networks." arXiv:2506.16114.
> >
> > [10] Yili Jin, Ling Pan, Rui-Xiao Zhang, Jiangchuan Liu, Xue Liu. "Generative Flow Networks for Personalized Multimedia Systems: A Case Study on Short Video Feeds." ACM Multimedia 2025.
>
> In this task, the agent is tasked with autoregressively generating a list (slate) of $K=6$ unique items from a large discrete action space of 3,706 items. We utilized a subset of the dataset containing **real user interaction trajectories** to simulate the offline setting, where intermediate states consist of the actual list of movies already watched and rated by the user, and a reward is assigned based on the average rating once $K=6$ items are rated. Crucially, since the quality of a recommendation list is invariant to the relative order of items, distinct generation orders lead to the same terminal set. Thus, this environment constitutes a **non-degenerate DAG** rather than a tree.
>
> The results are presented in the table below:
>
> | Method | Avg. Reward ($\uparrow$) | Max Reward ($\uparrow$) | Coverage ($\uparrow$) |
> | :--- | :---: | :---: | :---: |
> | Oracle-GFN | **2.092 $\pm$ 0.033** | **2.930 $\pm$ 0.017** | 113.7 $\pm$ 72.0 |
> | Dataset-GFN | 1.681 $\pm$ 0.202 | 2.674 $\pm$ 0.184 | 7.9 $\pm$ 0.9 |
> | QM-COFlowNet | 1.787 $\pm$ 0.022 | 2.843 $\pm$ 0.020 | 119.2 $\pm$ 6.1 |
> | **TD-GFN (Ours)** | **1.988 $\pm$ 0.006** | **2.914 $\pm$ 0.013** | **186.8 $\pm$ 6.3** |
>
> As shown, TD-GFN consistently outperforms other offline GFN baselines (Dataset-GFN and QM-COFlowNet). Notably, it achieves reward performance competitive with the Oracle-GFN which learning directly from the oracle environment, while exhibiting significantly higher and more stable diversity (Coverage).
>
> We have included these new experimental results and detailed settings in **Appendix D.1** of the revised paper. We envision that our approach can potentially unlock the potential for GFlowNet applications across a broader range of domains and scenarios.

---

> ### Author Response · Authors · 2025-11-20
> **Response to Reviewer cpmu (5/7)**
>
> ---
> > Weakness 4.2:
> > > - Seconds, the meaning of learning proxies from provided data is to obtain reward values over the entire terminal space; while imperfect, such proxies can facilitate exploration beyond $\mathcal{X}$ and discover new modes. This further underscores why capturing the overall terminal distributions matters[1].
> >
> > [1] Jain, Moksh, Tristan Deleu, Jason Hartford, Cheng-Hao Liu, Alex Hernandez-Garcia, and Yoshua Bengio. "Gflownets for ai-driven scientific discovery." Digital Discovery 2, no. 3 (2023): 557-577.
>
> **Response:**
>
> Thank you for your insightful comment regarding exploration. We appreciate this opportunity to clarify our specific to achive exploration.
>
> In fact, TD-GFN achieves exploration through a mechanism fundamentally different from proxy-based methods. By leveraging reward-guided pruning to prevent the policy from entering low-utility regions, and employing prioritized backward sampling to strategically allocate the model's attention based on rewards, we effectively achieve **exploration of unobserved high-utility regions during the evaluation phase**.
>
> Moreover, this mechanism exhibits a particularly strong synergy with the Flow Matching (FM) objective. Under FM, the flow function $F_\theta(s, s')$ effectively functions as **an implicit edge-level reward model**. From this perspective, our pruning method reshapes the learning landscape for this ''flow-reward'' model, simplifying its optimization task, while our prioritized backward sampling guides it to generalize strategically toward high-utility regions. **Analogous to how a proxy reward model generalizes across terminal states**, our approach enables this flow-reward model to generalize to unseen edges, thereby **actively guiding the policy to explore novel states**.
>
> We have added these points of discussion to the revised paper for better clarification and interpretation as follows:
>
> **In Section 3.2**
> > By leveraging reward-guided pruning to steer the policy away from low-utility regions, and employing prioritized backward sampling to strategically allocate the model’s attention based on rewards, we effectively achieve **exploration of unobserved high-utility regions during the evaluation phase**—a paradigm fundamentally distinct from the trial-and-error exploration inherent in proxy-based methods.
>
> **In Appendix D.8**
> > We hypothesize this is because the flow function $F_\theta(s, s')$ in the FM objective can itself be interpreted as an implicit edge-level reward model. From this perspective, our pruning method effectively reshapes the learning landscape for this 'flow-reward' model, simplifying its learning task, while our prioritized backward sampling guides it to strategically generalize to high-reward regions. Analogous to how a proxy reward model generalizes across terminal states, our approach enables this flow-reward model to generalize to unseen edges, thereby actively guiding the policy to explore novel states. This synergy might explain the FM objective's enhanced robustness and exploratory capability within our framework.

---

> ### Author Response · Authors · 2025-11-20
> **Response to Reviewer cpmu (6/7)**
>
> ---
> > Weakness 5: If the stated advantage of TD-GFN is better performance (in terms of either capturing distribution or discovering modes) than proxy-based GFNs, the evaluation does not establish this fairly. Most results compare against proxy-free GFN/RL baselines; rigorous baselines using conventional proxy-based GFlowNets (e.g., TB, Sub-TB with learned proxies) are absent or underreported. Moreover, because conventional proxy-based approaches are offline, there are substantial prior works on offline data samplers with replay buffers (e.g. [2]). As they does not touch the conventional training objectives, these GFN methods are clearly more straightforward than the proposed tree-phase pipeline ( IRL > DAG-pruning > policy updating) and should be included for a fair comparison.
> >
> > [2] Kim, Minsu, et al. "Local Search GFlowNets." The Twelfth International Conference on Learning Representations.
>
> **Response:**
>
> Thank you for this quetion. First, as elaborated in our response to Weakness 3, we do not claim that the proxy-free method is inherently superior to proxy-based approaches in general. Rather, our contribution is **complementary to proxy-based methods**, addressing the specific yet critical scenario **where training a reliable proxy is infeasible**. We have revised the Introduction to more clearly articulate this motivation. Furthermore, as demonstrated by the new results in Appendix D.12, naively applying conventional GFlowNet objectives directly to offline trajectories is insufficient for performance, underscoring the necessity of our proposed structural guidance.
>
> Regarding the specific suggestion to compare with data samplers utilizing replay buffers, we respectfully clarify that these **off-policy** GFlowNet training methods *cannot* be directly applied to the **offline** setting, as they fundamentally rely on the *online* collection of new trajectories to replenish the buffer, which necessitates a reward source (proxy) to evaluate these newly collected samples.
>
> Taking the suggested Local Search GFlowNets [2] as a specific example, we note that its direct application to the offline setting presents a fundamental challenge. Its core mechanism involves a "Refining" step where the policy generates a *new* trajectory $\tau'$ by locally modifying an existing one via backtracking and reconstruction. Crucially, to utilize this new trajectory, the algorithm explicitly **requires the evaluation of its reward** $R(\tau')$ for two indispensable purposes: first, to compute the acceptance probability for the local search step; and second, to label its ground-truth reward for subsequent training updates. In the offline setting, such reward feedback for novel, unobserved trajectories is unavailable.
>
> > [11] Minsu Kim, Taeyoung Yun, Emmanuel Bengio, Dinghuai Zhang, Yoshua Bengio, Sungsoo Ahn, Jinkyoo Park. "Local Search GFlowNets." ICLR 2024.
>
> ---
> > Weakness 6：While it is claimed by the authors, the paper does not demonstrate a true exploration–exploitation balance. Exploration (novel terminal-state visitation) and exploitation (concentration on known high-reward terminals) are not controlled. Because TD-GFN is proxy-free, it primarily optimizes a policy around the given limited set of rewarded terminals, i.e., exploitation.
>
> **Response:**
>
> Thank you for raising this critical point. However, we respectfully disagree with the characterization that our method, being proxy-free, *only* performs exploitation.
>
> The extensive diversity results in our experiments are strong evidence against the pure-exploitation claim. Our method consistently discovers a high number of diverse, high-reward modes, far exceeding the limited set of terminals in the training data. This directly demonstrates that our method is not confined to optimizing solely around the limited set of rewarded terminals found in the training data.
>
> As elaborated in our response to Weakness 4.2, this capability stems from our unique training mechanism. By strategically directing the policy toward high-utility regions via reward-guided pruning and prioritized sampling, we effectively achieve exploration during the evaluation phase. Furthermore, the learned flow function in the Flow Matching objective acts as an implicit edge-level reward model that generalizes to unseen transitions—analogous to how a proxy generalizes across terminal states—thereby actively guiding the policy to explore novel states.
>
> We have added these points of discussion to **Section 3.2 and Appendix D.8** in the revised paper for better clarification and interpretation, as shown in our response to Weakness 4.2. Thanks for your constrctive feedback!

---

> ### Author Response · Authors · 2025-11-20
> **Response to Reviewer cpmu (7/7)**
>
> ---
> > Question 2: Why the training curves of AMP tasks are not provided?
>
> **Response:**
>
> We have provided the training curves for the AMP task in the **Appendix D.11** of the revised paper. We initially omitted them because, while **TD-GFN remains stable throughout training**, baseline methods exhibit significant volatility. To ensure a fair comparison in the main text, we reported the baseline results at their best-performing checkpoints.
>
> ---
> > Question 3: Where is the diversity reporting for sEH task?
>
> **Response:**
>
> Thank you for your question! We computed the average internal Tanimoto similarity among the top-100 highest-reward molecules sampled from the final policies of each method.
>
> The results are as follows:
>
> | Method | Top-100 Internal Tanimoto Sim. ($\downarrow$) | Top-100 Reward ($\uparrow$) |
> | :--- | :---: | :---: |
> | Proxy-GFN | 0.665 $\pm$ 0.024 | 7.281 $\pm$ 0.067 |
> | Oracle-GFN | 0.615 $\pm$ 0.017 | 7.408 $\pm$ 0.021 |
> | Dataset-GFN | 0.521 $\pm$ 0.026 | 7.198 $\pm$ 0.018 |
> | FM-COFlowNet | 0.535 $\pm$ 0.015 | 7.201 $\pm$ 0.015 |
> | QM-COFlowNet | 0.526 $\pm$ 0.014 | 7.296 $\pm$ 0.022 |
> | **TD-GFN (Ours)** | 0.531 $\pm$ 0.022 | **7.450 $\pm$ 0.037** |
>
> As this table shows, our algorithm does not sacrifice diversity while ensuring sampling optimality. This explains why, as shown in our main experiment on the Molecule Design task, our method is able to discover a larger number of high-reward modes. We have added these results to **Appendix D.10** in the revised paper for reference.
>
> ---
> We hope our response addresses your concerns. If so, we wonder if you could kindly consider raising your score? We will also be happy to answer any further questions you may have. Thank you very much!

---

### Official Review · Reviewer_f6if · 2025-11-01

**Soundness:** 3
**Presentation:** 3
**Contribution:** 3
**Rating:** 8
**Confidence:** 3

**Summary:**

This paper introduces Trajectory-Distilled GFlowNet (TD-GFN), a proxy-free framework for training Generative Flow Networks on offline datasets. To overcome limitations of existing methods—such as error propagation in proxy-based approaches and restricted exploration in proxy-free methods—TD-GFN employs inverse reinforcement learning to derive fine-grained edge rewards from trajectories. These rewards indirectly guide policy learning via directed acyclic graph pruning and prioritized backward sampling, ensuring updates rely only on ground-truth terminal rewards. Experiments demonstrate that TD-GFN achieves superior convergence speed and sample quality, offering a more robust and efficient paradigm for offline GFlowNet training.

**Strengths:**

**Novel Proxy-Free Paradigm**: Departing from existing paradigms, this work pioneers a proxy-free approach that leverages estimated edge rewards. This novel framework effectively circumvents the key limitations inherent in both proxy-based methods, such as error propagation, and prior proxy-free methods, which often rely on coarse-grained constraints.

**High Efficiency**: By integrating DAG pruning and prioritized backward sampling, TD-GFN sets a new state of the art for offline GFlowNet training. It demonstrates robust superiority over a wide range of baselines, achieving faster convergence, higher sample quality, and a closer fit to the target distribution.

**Clear Presentation and Comprehensive Results**: The paper is well-written with clear motivation and methodology, and the appendix is very comprehensive. The contribution of the paper looks solid.

**Weaknesses:**

**Insufficient Analysis for Design Choices** : Although this paper does a pretty good ablation study in the appendix, it lacks analysis on the detailed design choices and what problems this design might lead to. For instance, pruning seems like a more 'extreme' version of weighted sampling, so would it also work to remove this part while somewhat adjusting the weighted sampling method to make it 'harsher' for those low-reward edges? In practice, purely clean datasets are often difficult to obtain in real-world scenarios, and pruning seems to make it prone to mistakes that removes some actually important edges. It is just an example, but I believe if the authors have tried more options and finally chosen this design, there should have been more details and insights here.

**Limited Experiment Settings**: The experiment results are mostly based on synthetic datasets, while only the biosequences are real-world data. Moreover, 1500 trajectories are relatively small, which does not prove if the proposed method has similar scalabilities as baselines.

**Insufficient Baselines**: It would also be beneficial if they can also compare the results with some non-GFlownet methods, i.e. some recent offline-RL methods.

**Questions:**

Just what I've mentioned in the 'weakness' part.

---

> ### Author Response · Authors · 2025-11-20
> **Response to Reviewer f6if (1/3)**
>
> Thank you for your time and effort in reviewing our paper! We are very grateful for your high score and for recognizing the **novelty** of our proxy-free paradigm, the resulting **high efficiency** in GFlowNet training, and the **clarity and solidity** of our contribution.
>
> We try our best to address your questions as follows. Please note that all revisions in the updated manuscript are highlighted in **blue** for your convenience.
>
> ---
> > Weakness 1: **Insufficient Analysis for Design Choices :** Although this paper does a pretty good ablation study in the appendix, it lacks analysis on the detailed design choices and what problems this design might lead to. For instance, pruning seems like a more 'extreme' version of weighted sampling, so would it also work to remove this part while somewhat adjusting the weighted sampling method to make it 'harsher' for those low-reward edges? In practice, purely clean datasets are often difficult to obtain in real-world scenarios, and pruning seems to make it prone to mistakes that removes some actually important edges. It is just an example, but I believe if the authors have tried more options and finally chosen this design, there should have been more details and insights here.
>
> **Response:**
>
> Thank you for this insightful question. Your concern about pruning being an 'extreme' measure that might be 'prone to mistakes' is perfectly valid. Our design was, in fact, **chosen specifically to be robust against the potential numerical errors in the learned IRL rewards**.
>
> In fact, we explored using a 'soft' guidance mechanism by re-weighting the policy logits using the learned edge rewards. However, we found that the weighting strength became a **sensitive** hyperparameter that required careful tuning to achieve performance comparable to pruning on the Hypergrid distribution-matching task.
>
> A potential explanation is that rewards learned via IRL, while directionally useful, contain **numerical generalization errors**, which may lead to sub-optimal performance. Our current threshold-pruning method, in contract, only requires the IRL model to be sufficient at the task of **distinguishing low-reward edges from potentially useful ones**, mitigating the impact of such numerical errors.
>
> Our analyses confirm that this design is robust. **Appendix C.4 and D.5** show that this approach is **robust with respect to its hyperparameter (the pruning threshold)**. Our main experiments in **Section 4.1 and Appendix D.2** demonstrate that the entire TD-GFN framework is highly robust to dataset uncertainties, showing strong performance on noisy ('Mixed', 'Bad') and sparse ('Mixed-150', 'Mixed-30') datasets where distinguishing signal from noise is critical.
>
> We look forward to future work finding more advanced 'soft' methods, as we noted in the Appendix E of our initial submission. We have clarified the rationale for our pruning choice in the Methodology section and incorporated these observations from our design exploration into the revised paper to provide more experience for future research, as follows:
>
> **In Section 3.2:**
> > By adopting this threshold-based pruning strategy, we rely primarily on the model’s ability to distinguish low-reward edges from potentially useful ones, thereby mitigating the risk of policy degradation caused by potential numerical errors in the learned IRL rewards. We further discuss alternative “soft” guidance mechanisms in Appendix E.
>
> **In Appendix E "future directions" paragragh:**
>
> > Furthermore, an interesting future direction would be to replace the “hard” pruning of the DAG with a “softer” guidance mechanism. In fact, the pruning method can be formally understood as an extreme case of a “soft” guidance mechanism, such as re-weighting the policy logits using the learned edge rewards. In this view, pruning is equivalent to applying an infinitely large positive weight to transitions above the threshold and a zero weight (or infinitely negative logit) to those below it. This perspective provides a clear theoretical motivation, as this re-weighting can be interpreted as imposing a strong Bayesian prior on the policy (incorporating the structural knowledge learned via IRL) or as a form of imitation learning regularization that constrains the policy to align with the distilled expert behaviors.
>
> >However, based on our initial explorations, while this “soft” method is a compelling alternative, it seems to be highly sensitive to its weighting-strength hyperparameter. A potential explanation is that it relies on the precise numerical accuracy of the learned edge rewards, which may contain generalization errors. Developing a more effective “soft” guidance mechanism remains a promising avenue for future research.

---

> > ### Author Response · Authors · 2025-11-20
> > **Response to Reviewer f6if (2/3)**
> >
> > ---
> > > Weakness 2.1: **Limited Experiment Settings:** The experiment results are mostly based on synthetic datasets, while only the biosequences are real-world data.
> >
> > **Response:**
> >
> > Thank you for this valuable suggestion! We have incorporated an additional experiment on the Listwise Recommendation task using the MovieLens 1M (ML1M) dataset [1], a standard benchmark containing real user-item interactions.
> >
> > > [1] Shuchang Liu, Qingpeng Cai, Zhankui He, Bowen Sun, Julian McAuley, Dong Zheng, Peng Jiang, Kun Gai. "Generative Flow Network for Listwise Recommendation." ACM KDD 2023.
> >
> > * **Task Definition:** The agent is tasked with autoregressively generating a list (slate) of $K=6$ unique items from a large discrete action space of 3,706 items.
> > * **Real-World Trajectories:** We utilized a subset of the dataset containing **real user interaction trajectories** to simulate the offline setting. Intermediate states consist of the list of movies already watched and rated by the user. A reward is assigned once $K=6$ items are rated, calculated as the average of the user's ratings.
> > * **Non-Degenerate DAG Structure:** Crucially, since the quality of a recommendation list is invariant to the relative order of items, this environment constitutes a non-degenerate DAG rather than a tree, as different generation orders lead to the same terminal set.
> >
> > The results are presented in the table below:
> >
> > | Method | Avg. Reward ($\uparrow$) | Max Reward ($\uparrow$) | Coverage ($\uparrow$) |
> > | :--- | :---: | :---: | :---: |
> > | Oracle-GFN | **2.092 $\pm$ 0.033** | **2.930 $\pm$ 0.017** | 113.7 $\pm$ 72.0 |
> > | Dataset-GFN | 1.681 $\pm$ 0.202 | 2.674 $\pm$ 0.184 | 7.9 $\pm$ 0.9 |
> > | QM-COFlowNet | 1.787 $\pm$ 0.022 | 2.843 $\pm$ 0.020 | 119.2 $\pm$ 6.1 |
> > | **TD-GFN (Ours)** | **1.988 $\pm$ 0.006** | **2.914 $\pm$ 0.013** | **186.8 $\pm$ 6.3** |
> >
> > As shown, TD-GFN consistently outperforms other offline GFN baselines (Dataset-GFN and QM-COFlowNet). Notably, it achieves reward performance competitive with the Oracle-GFN, which learning directly from the oracle environment, while exhibiting significantly higher and more stable diversity (Coverage). We have added these experimental results and the detailed experimental settings to **Appendix D.1** of the revised paper.
> >
> > ---
> > > Weakness 2.2: Moreover, 1500 trajectories are relatively small, which does not prove if the proposed method has similar scalabilities as baselines.
> >
> > **Response:**
> >
> > Thanks a lot for your noticing!
> >
> > First, in the Hypergrid environment (Section 4.1 and Appendix D.2), our experiments across varying dataset sizes (30, 150, and 1,500 trajectories) demonstrate a stable and monotonic improvement in performance. Notably, at 1,500 trajectories, the algorithm's performance approaches saturation—it rapidly identifies all modes and matches the target distribution with minimal L1 error. This indicates that 1,500 trajectories are sufficient for TD-GFN to efficiently solve this complex task, rather than indicating a scalability limit.
> >
> > Second, in the Molecule Design task, the use of 1,500 sub-optimal trajectories was a deliberate choice to highlight the **superior data efficiency** of our approach. As detailed in Section 4.3, the Oracle-GFN baseline relies on an oracle reward model pre-trained on a massive dataset of 300k molecules. The fact that TD-GFN matches the performance of this oracle-guided policy using only 1,500 trajectories demonstrates its capability to distill high-quality guidance without requiring extensive pre-training data .
> >
> > We believe these results confirm that TD-GFN opens new possibilities for applying GFlowNets in data-constrained scenarios.

---

> ### Author Response · Authors · 2025-11-20
> **Response to Reviewer f6if (3/3)**
>
> ---
> > Weakness 3: **Insufficient Baselines:** It would also be beneficial if they can also compare the results with some non-GFlownet methods, i.e. some recent offline-RL methods.
>
> **Response:**
>
> This is an excellent suggestion. Following your advice, we implemented BraVE [2], a state-of-the-art offline RL algorithm specifically designed for discrete combinatorial action spaces, on the Molecule Design task using its official implementation.
>
> > [2] Matthew Landers, Taylor W. Killian, Hugo Barnes, Thomas Hartvigsen, Afsaneh Doryab. "BraVE: Offline Reinforcement Learning for Discrete Combinatorial Action Spaces." NeurIPS 2025.
>
> | Method | Reward-10 ($\uparrow$) | Reward-100 ($\uparrow$) | Reward-1000 ($\uparrow$) | Convergence ($\downarrow$) |
> | :--- | :---: | :---: | :---: | :---: |
> | CQL | 7.069 | 6.643 | 5.401 | 0.803 $\times 10^4$ |
> | IQL | 6.902 | 5.980 | 4.628 | 4.104 $\times 10^4$ |
> | BraVE | 7.271 | 6.650 | 5.590 | **0.645 $\mathbf{\times 10^4}$** |
> | **TD-GFN (Ours)** | **7.733** | **7.450** | **6.810** | **2.749 $\mathbf{\times 10^4}$** |
>
> While BraVE demonstrates clear improvements over standard offline RL baselines like CQL and IQL, our results indicate that it still struggles with the fundamental misalignment between the standard RL paradigm (which focuses on reward maximization) and the objective of this task (which requires diverse sampling). Specifically, while offline RL methods can efficiently locate several high-reward modes in sparse reward landscapes—as evidenced by BraVE's fast convergence—they tend to collapse into this local optimum. Consequently, they fail to sample the diverse set of high-reward modes that GFlowNets are designed to discover.
>
> We believe our TD-GFN framework effectively bridges this gap, offering a robust offline GFlowNet paradigm that succeeds where pure RL methods fall short in diversity-centric tasks.
>
> We have added these results and the accompanying analysis to the main experiment (**Section 4.3**) in the revised paper.
>
> ---
> We hope our response addresses your concerns. If so, we wonder if you could kindly consider raising your confidence? We will also be happy to answer any further questions you may have. Thank you very much!

---

### Author Response · Authors · 2025-11-27
**General Response: Summary of Updates and New Experiments**

We thank all reviewers for their time and the constructive feedback provided during the discussion period. Your insights have significantly helped us strengthen our empirical evaluation and clarify the positioning of our work.

As the discussion period progresses, we would like to take this opportunity to summarize the major updates we have incorporated into the revised manuscript (**changes highlighted in blue**) to address the common and specific concerns raised.

---
**1. New Real-World Trajectory Experiment (Addressing Reviewers DWcy, f6if, cpmu)**

To address the concern regarding the reliance on synthetic data and demonstrate the applicability of TD-GFN to **real-world, non-degenerate (trajectory-level) DAGs**, we have added a Listwise Recommendation task using the MovieLens 1M dataset (real user interaction trajectories) in **Appendix D.1**.
* **Result:** TD-GFN significantly outperforms offline GFN baselines (Dataset-GFN, QM-COFlowNet) and achieves reward performance competitive with the Oracle-GFN while maintaining significantly higher diversity.

**2. New Baselines and Comparisons (Addressing Reviewers f6if, cpmu)**
* **Advanced Offline RL:** We implemented **BraVE** (NeurIPS 2025), a specialized offline RL method for discrete combinatorial spaces, on the Molecule Design task in **Section 4.3**. The results highlight that while BraVE improves over standard RL baselines (CQL/IQL) in convergence, TD-GFN still demonstrates superior diversity and optimality.
* **Naive GFN Objectives:** We evaluated standard GFlowNet objectives (TB, Sub-TB, DB) applied directly to offline trajectories in **Appendix D.12**. Results confirm that naive application fails without the structural guidance provided by our framework.

**3. Scalability to Long Horizons (Addressing Reviewer cpmu)**
We conducted additional experiments on a **$256^2$ Hypergrid** in **Appendix D.3**.
* **Result:** TD-GFN successfully identifies **all** 4 widely separated reward modes using only 100 training trajectories, validating the efficacy of our pruning and edge-reward guidance in long-horizon tasks.

**4. Enhanced Diversity Analysis (Addressing Reviewer cpmu)**

We added a post-hoc analysis of **pairwise internal Tanimoto similarity** for the Molecule Design task in **Appendix D.10**.
* **Result:** The analysis confirms that TD-GFN achieves higher rewards without sacrificing sample diversity compared to baselines.

**5. Deeper Theoretical and Design Discussions (Addressing Reviewers DWcy, f6if, i231)**
* **Pruning vs. Soft Guidance:** We expanded the discussion in **Section 3.2** and **Appendix E** to explain the theoretical motivation behind our "hard" pruning strategy (robustness against numerical IRL errors) compared to soft" re-weighting mechanisms.
* **IRL vs. Imitation:** We clarified why the discriminator-based IRL formulation is necessary for generalizing to out-of-distribution regions (**Section 3.1**).
* **Exploration Mechanism:** We clarified how our method achieves exploration during the evaluation phase via reward-guided pruning and prioritized backward sampling, leveraging the implicit reward modeling of Flow Matching (**Section 3.2**, **Appendix D.8**).

**6. Clarified Motivation and Positioning (Addressing Reviewer cpmu)**

We have revised the **Abstract** and **Introduction** to explicitly clarify our research scope. We emphasize that our work addresses scenarios where **constructing a reliable proxy is infeasible due to data scarcity or cost**. We further clarified in **Appendix E** that comparisons with Oracle-GFN serve to benchmark the gap between offline training and ideal online settings rather than implying direct competition between proxy-based methods and our method.

---
We have responded to each reviewer's specific questions in the individual threads. We will be happy to answer any further questions in the remaining time.

Best regards,
The Authors

---

### Meta-Review · Area_Chair_9MMG · 2026-01-07

**Summary:**

The authors presented a "proxy-free" Trajectory-Distilled GFlowNet (TD-GFN) training, which adopts inverse reinforcement learning (GAIL) to derive edge rewards (as a discriminator) from offline trajectories. With edge pruning, DG-GFN is claimed to better guide GFlowNet sampling/generation to achieve superior sample quality with faster convergence.

**Reviewer Concerns:**

Based on available reviewer-author discussions, the authors have tried to clarify offline proxy-free training of TD-GRN and several design choices, including the use of edge discriminator and edge pruning.

All reviewers requested additional experiments, either more baseline benchmarking or more task examples. The authors have tried to provide these additional results.

One of the remaining concerns, especially from reviewer cpmu is the potential issue on evaluation. The reviewer raised the question whether convergence comparison is fair, especially in Figure 1. As the proposed TD-GFN converges much faster than Oracle-GFN, the authors may need to discuss carefully on this due to the additional trajectories have been used to train TD-GFN offline, especially considering the training and sampling settings follow the similar underlying transition and reward functions for these settings. It is not fair for such convergence comparison without explicitly stating this potential bias. Secondly, top-k reward and mode evaluation metrics are also not fair evaluation metrics for GFlowNet sampling quality (as the authors recognized in the rebuttal, they are not targeting at distribution learning though), since they may have issues if TD-GFN keeps sampling regions around the same modes. The authors may want to consider evaluate the mode coverage or the number of discovered modes besides the presented metrics including diversity checking, which may also better address other reviewers (e.g. i231)' concerns.

This reviewer (cpmu) also requested benchmarking with "Local Search GFlowNet" but the authors stated that there are fundamental challenge in implementing it. Based on my understanding, the reviewer asked for similar benchmarking as the authors presented in Appendix D.12 (Figure 20) for DB, TB based GFlowNet training.

Reviewer i231's suggestion for integration into online setting may fall along the same direction. The authors stated that that will be their future research direction.

Reviewer DWcy requested for "real theoretical explanation or motivation" but the provided explanations and revision simply explained the claimed benefits coming from "Bayesian prior" motivation by TD-GFN, which may not directly answer the reviewer's questions.

**Reviewer Scores:**

Based on the available discussions, the reviewers may not be able to reach the consensus (especially reviewer cpmu, possibly DWcy) for acceptance recommendation. The authors may consider better positioning the paper and adding more evaluation metrics for fair and convincing evaluation.

---

### Decision · Program_Chairs · 2026-01-26

Reject